# First-order methods almost always avoid saddle points: The case of vanishing step-sizes

**Ioannis Panageas**
SUTD
Singapore
ioannis@sutd.edu.sg

**Georgios Piliouras**
SUTD
Singapore
georgios@sutd.edu.sg

**Xiao Wang**
SUTD
Singapore
xiao_wang@sutd.edu.sg

## Abstract

In a series of papers [17, 22, 16], it was established that some of the most commonly used first order methods almost surely (under random initializations) and with step-size being small enough, avoid strict saddle points, as long as the objective function $f$ is $C^2$ and has Lipschitz gradient. The key observation was that first order methods can be studied from a dynamical systems perspective, in which instantiations of Center-Stable manifold theorem allow for a global analysis. The results of the aforementioned papers were limited to the case where the step-size $\alpha$ is constant, i.e., does not depend on time (and bounded from the inverse of the Lipschitz constant of the gradient of $f$). It remains an open question whether or not the results still hold when the step-size is time dependent and vanishes with time.

In this paper, we resolve this question on the affirmative for gradient descent, mirror descent, manifold descent and proximal point. The main technical challenge is that the induced (from each first order method) dynamical system is time non-homogeneous and the stable manifold theorem is not applicable in its classic form. By exploiting the dynamical systems structure of the aforementioned first order methods, we are able to prove a stable manifold theorem that is applicable to time non-homogeneous dynamical systems and generalize the results in [16] for vanishing step-sizes.

## 1 Introduction

Non-convex optimization has been studied extensively the last years and has been one of the main focuses of Machine Learning community. The reason behind the interest of ML community is that in many applications of interest, one has to deal with the optimization of a non-convex landscape. One of the key obstacles of non-convex optimization is the existence of numerous saddle points (which can outnumber the local minima [10, 24, 6]). Avoiding them is a fundamental challenge for ML [14].

Recent progress [11, 16] has shown that under mild regularity assumptions on the objective function, first-order methods such as gradient descent can provably avoid the so-called *strict* saddle points[1].

In particular, a unified theoretical framework is established in [16] to analyze the asymptotic behavior of first-order optimization algorithms such as gradient descent, mirror descent, proximal point, coordinate descent and manifold descent. It is shown that under random initialization, the aforementioned methods avoid strict saddle points almost surely. The proof exploits a powerful theorem from the dynamical systems literature, the so-called Stable-manifold theorem (see supplementary material for a statement of this theorem). For example, given a $C^2$ (twice continuously differentiable function) $f$ with $L$-Lipschitz gradient, gradient descent method

$$\mathbf{x}_{k+1} = g(\mathbf{x}_k) := \mathbf{x}_k - \alpha \nabla f(\mathbf{x}_k)$$

avoids strict saddle points almost surely, under the assumption that the stepsize is *constant* and $0 < \alpha < \frac{1}{L}$. The crux of the proof in [16] is the Stable-manifold theorem for the *time-homogeneous*[2] dynamical system $\mathbf{x}_{k+1} = g(\mathbf{x}_k)$. The Stable-manifold theorem implies that the dynamical system $g$ avoids its unstable fixed points and with the fact that the unstable fixed points of the dynamical system $g$ coincide with the strict saddles of $f$ the claim follows. Results of similar flavor can be shown for Expectation Maximization algorithm [19], Multiplicative Weights Update [18, 23] and for min-max optimization [9].

In many applications/algorithms, however, the stepsize is adaptive or vanishing/diminishing (meaning $\lim_k \alpha_k = 0$, e.g., $\alpha_k = \frac{1}{k}$ or $\frac{1}{\sqrt{k}}$). Such applications include stochastic gradient descent (see [27] for analysis of SGD for convex functions), urn models and stochastic approximation [25], gradient descent [4], online learning algorithms like multiplicative weights update [1, 15] (which is an instantiation of Mirror Descent with entropic regularizer). It is also important to note that the choice of the stepsize is really crucial in the aforementioned applications as changing the stepsize can change the convergence properties (transition from convergence to oscillations/chaos [20, 21, 8, 5]), the rate of convergence [20] as well as the system efficiency [7].

The proof in [16] does not carry over when the stepsize depends on time, because the Stable-manifold theorem is not applicable. Hence, whether the results of [16] hold for vanishing step-sizes remains unresovled. This was stated explicitly as an open question in [16]. Our work resolves this question in the affirmative. Our main result is stated below informally.

**Theorem 1.1** (Informal)**.** *Gradient Descent, Mirror Descent, Proximal point and Manifold descent, with vanishing step-size $\alpha_k$ of order $\Omega\left(\frac{1}{k}\right)$ avoid the set of strict saddle points (isolated and non-isolated) almost surely under random initialization.*

**Organization of the paper.**    The paper is organized as follows: In Section 2 we give important definitions for the rest of the paper, in Section 3 we provide intuition and technical overview of our results, in Section 4 we show a new Stable-manifold theorem that is applicable to a class of time non-homogeneous dynamical systems and finally in Section 5 we show how this new manifold theorem can be applied to Gradient descent, Mirror Descent, Proximal point and Manifold Descent. Due to space constraints, most of the proofs can be found in the supplementary material.

**Notation.**    Throughout this paper, we denote $\mathbb{N}$ the set of nonnegative integers and $\mathbb{R}$ the set of real numbers, $\|\cdot\|$ the Euclidean norm, bolded $\mathbf{x}$ the vector, $\mathbb{B}(\mathbf{x}, \delta)$ the open ball centering at $\mathbf{x}$ with radius $\delta$, $g(k, \mathbf{x})$ the update rule for optimization algorithms indexed by $k \in \mathbb{N}$, $\tilde{g}(m, n, \mathbf{x})$ the composition $g(m, ..., g(n+1, g(n, \mathbf{x}))...)$ for $m \geq n$, $\nabla f$ the gradient of $f : \mathbb{R}^d \to \mathbb{R}$ and $\nabla^2 f(\mathbf{x})$ the Hessian of $f$ at $\mathbf{x}$, $D_{\mathbf{x}} g(k, \mathbf{x})$ the differential with respect to variable $\mathbf{x}$,

## 2   Preliminaries

In this section we provide all necessary definitions that will be needed for the rest of the paper.

**Definition 2.1** (Time (non)-homogeneous)**.** *We call a dynamical system $x_{k+1} = g(x_k)$ as time homogeneous since $g$ does not on step $k$. Furthermore, we call a dynamical system $x_{k+1} = g(k, x_k)$ time non-homogeneous as $g$ depends on $k$.*

**Definition 2.2** (Critical point)**.** *Given a $C^2$ (twice continuously differentiable) function $f : \mathbb{X} \to \mathbb{R}$ where $\mathbb{X}$ is an open, convex subset of $\mathbb{R}^d$, the following definitions are provided for completeness.*

1. *A point $\mathbf{x}^*$ is a critical point of $f$ if $\nabla f(\mathbf{x}^*) = 0$.*

2. *A critical point is a local minimum if there is a neighborhood $U$ around $\mathbf{x}^*$ such that $f(\mathbf{x}^*) \leq f(\mathbf{x})$ for all $\mathbf{x} \in U$, and a local maximum if $f(\mathbf{x}^*) \geq f(\mathbf{x})$.*

3. *A critical point is a saddle point if for all neighborhoods $U$ around $\mathbf{x}^*$, there are $\mathbf{x}, \mathbf{y} \in U$ such that $f(\mathbf{x}) \leq f(\mathbf{x}^*) \leq f(\mathbf{y})$.*

4. *A critical point $\mathbf{x}^*$ is isolated if there is a neighborhood $U$ around $\mathbf{x}^*$, and $\mathbf{x}^*$ is the only critical point in $U$.*

This paper will focus on saddle points that have directions of strictly negative curvature, that is the concept of strict saddle.

**Definition 2.3** (Strict Saddle). *A critical point $\mathbf{x}^*$ of $f$ is a strict saddle if $\lambda_{\min}(\nabla^2 f(\mathbf{x}^*)) < 0$ (minimum eigenvalue of the Hessian computed at the critical point is negative).*

Let $\mathcal{X}^*$ be the set of strict saddle points of function $f$ and we follow the Definition 2 of [16] for the global stable set of $\mathcal{X}^*$.

**Definition 2.4** (Global Stable Set and fixed points). *Given a dynamical system (e.g., gradient descent $\mathbf{x}_{k+1} = \mathbf{x}_k - \alpha_k \nabla f(\mathbf{x}_k)$)*

$$\mathbf{x}_{k+1} = g(k, \mathbf{x}_k), \tag{1}$$

*the global stable set $W^s(\mathcal{X}^*)$ of $\mathcal{X}^*$ is the set of initial conditions where the sequence $\mathbf{x}_k$ converges to a strict saddle. This is defined as:*

$$W^s(\mathcal{X}^*) = \{\mathbf{x}_0 : \lim_{k \to \infty} \mathbf{x}_k \in \mathcal{X}^*\}.$$

*Moreover, $\mathbf{z}$ is called a fixed point of the system (1) if $\mathbf{z} = g(k, \mathbf{z})$ for all natural numbers $k$.*

**Definition 2.5** (Manifold). *A $C^k$-differentiable, $d$-dimensional manifold is a topological space $M$, together with a collection of charts $\{(U_\alpha, \phi_\alpha)\}$, where each $\phi_\alpha$ is a $C^k$-diffeomorphism from an open subset $U_\alpha \subset M$ to $\mathbb{R}^d$. The charts are compatible in the sense that, whenever $U_\alpha \cap U_\beta \neq \emptyset$, the transition map $\phi_\alpha \circ \phi_\beta^{-1} : \phi_\beta(U_\beta \cap U_\alpha) \to \mathbb{R}^d$ is of $C^k$.*

## 3 Intuition and Overview

In this section we will illustrate why gradient descent and related first-order methods do not converge to saddle points, even for time varying/vanishing step-sizes $\alpha_k$ of order $\Omega\left(\frac{1}{k}\right)$.

### 3.1 Intuition

Consider the case of a quadratic, $f(\mathbf{x}) = \frac{1}{2}\mathbf{x}^T A \mathbf{x}$ where $A = \mathbf{diag}(\lambda_1, ..., \lambda_d)$ is a $d \times d$, non-singular, diagonal matrix with at least a negative eigenvalue. Let $\lambda_1, ..., \lambda_j$ be the positive eigenvalues of $A$ (the first $j$) and $\lambda_{j+1}, ..., \lambda_d$ be the non-positive ones. It is clear that $\mathbf{x}^* = \mathbf{0}$ is the unique critical point of function $f$ and the Hessian $\nabla^2 f$ is $A$ everywhere (and hence at the critical point). Moreover, it is clear that $\mathbf{x}^*$ is a strict saddle point (not a local minimum).

Gradient descent with step-size $\alpha_k$ (it holds that $\alpha_k \geq 0$ for all $k$ and $\lim_{k \to \infty} \alpha_k = 0$) has the following form:

$$\mathbf{x}_{k+1} = \mathbf{x}_k + \alpha_k A \mathbf{x}_k = (I - \alpha_k A)\mathbf{x}_k.$$

Assuming that $\mathbf{x}_0$ is the starting point, then it holds that $\mathbf{x}_{k+1} = \left(\prod_{t=0}^{k}(I - \alpha_{k-t}A)\right)\mathbf{x}_0$. We conclude that

$$\mathbf{x}_{k+1} = \mathbf{diag}\left(\prod_{t=0}^{k}(1 - \lambda_1 \alpha_t), ..., \prod_{t=0}^{k}(1 - \lambda_n \alpha_t)\right)\mathbf{x}_0. \tag{2}$$

We examine when it is true that $\lim_{k \to \infty} \mathbf{x}_k = \mathbf{x}^*$. It is clear that $\prod_{t=0}^{\infty}(1 - \lambda \alpha_t) = e^{\sum_{t=0}^{\infty} \ln(1 - \lambda \alpha_t)}$, and has the same convergence properties as

$$e^{-\lambda \sum_{t=0}^{\infty} \alpha_t}. \tag{3}$$

For $\lambda > 0$, the term (3) converges to zero if and only if $\sum_{t=0}^{\infty} \alpha_t = +\infty$ which is true if $\alpha_t$ is $\Omega\left(\frac{1}{t}\right)$. Moreover, for $\lambda = 0$ it holds that the term (3) remains a constant (independently of the choice of stepsize $\alpha_k$) and for $\lambda < 0$ it holds that the term (3) diverges for $\alpha_t$ to be $\Omega\left(\frac{1}{t}\right)$. Therefore, for $\alpha_k$ being $\Omega\left(\frac{1}{k}\right)$ we conclude that $\lim_{k \to \infty} x_k = \mathbf{0}$ whenever the initial point $\mathbf{x}_0$ satisfies $x_0^i = 0$ ($i$-th coordinate of $\mathbf{x}_0$) for $\lambda_i \geq 0$.

Hence, if $\mathbf{x}_0 \in E_s := \mathrm{span}(e_1, \ldots, e_j)^3$, then $\mathbf{x}_t$ converges to the saddle point $x^*$ and if $\mathbf{x}_0$ has a component outside $E_s$ then gradient descent diverges. For the example above, the global stable set of $\mathbf{x}^*$ is the subspace $E_s$ which is of measure zero since $E_s$ is not full dimensional.

**Remark 3.1** ($\alpha_k$ of order $O\left(\frac{1}{k^{1+\epsilon}}\right)$). *In the case where $\alpha_k$ is a sequence of stepsizes that converges to zero with a rate $\frac{1}{k^{1+\epsilon}}$ for any $\epsilon > 0$ (example $\frac{1}{k^2}, \frac{1}{2^k}$ etc), then it holds that $\sum_{t=0}^{\infty} \alpha_k$ converges and hence in our example above we conclude that $\lim_{k \to \infty} \mathbf{x}_k$ exists, i.e., $\mathbf{x}_k$ converges but not necessarily to a critical point.*

## 3.2 Technical Overview

The stability of non-homogeneous (i.e. non-autonomous) systems, at least for the case of continuous-time systems, has been the subject of intensive investigation ([2] and references therein). Although some work on discrete-time systems exists [26], this area is less developed and as far as we know no explicit connections to optimization applications have been made before. Moreover, as far as gradient descent, mirror descent, etc are concerned, the corresponding dynamical system that needs to be analyzed is more complicated when the objective function is not quadratic and the analysis of previous subsection does not apply.

Suppose we are given a function $f$ that is $C^2$, and $\mathbf{0}$ is a saddle point of $f$. The Taylor expansion of the gradient descent in a neighborhood of $\mathbf{0}$ is as follows:

$$\mathbf{x}_{k+1} = (I - \alpha_k \nabla^2 f(\mathbf{0})) \mathbf{x}_k + \eta(k, \mathbf{x}_k), \tag{4}$$

where $\eta(k, \mathbf{0}) = \mathbf{0}$ and $\eta(k, \mathbf{x})$ is of order $o(\|\mathbf{x}\|)$ around $\mathbf{0}$ for all naturals $k$.

Due to the error term $\eta(k, \mathbf{x}_k)$, the approach for quadratic functions does not imply the existence of the stable manifold. Inspired by the proof of Stable-manifold theorem for time homogeneous ODEs, we prove a Stable-manifold theorem for discrete time non-homogeneous dynamical system (4). In words, we prove the existence of a manifold $W^s$ that is not of full dimension (it has the same dimension as $E^s$, where $E^s$ denotes the subspace that is spanned by the eigenvectors with corresponding positive eigenvalues of matrix $\nabla^2 f(\mathbf{0})$).

To show this, we derive the expression of (2) for the general function $f$ to be:

$$\mathbf{x}_{k+1} = A(k, 0) \mathbf{x}_0 + \sum_{i=0}^{k} A(k, i+1) \eta(i, \mathbf{x}_i), \tag{5}$$

where $A(m, n) = \left(I - \alpha_m \nabla^2 f(\mathbf{0})\right) \ldots \left(I - \alpha_n \nabla^2 f(\mathbf{0})\right)$ for $m \geq n$, and $A(m, n) = I$ if $m < n$. Next, we generate a sequence $\{\mathbf{x}_k\}_{k \in \mathbb{N}}$ from (5) with an initial point $\mathbf{x}_0 = (\mathbf{x}_0^+, \mathbf{x}_0^-)$, where $\mathbf{x}_0^+ \in E^s$ and $\mathbf{x}_0^- \in E^u$. If this sequence converges to $\mathbf{0}$, the equation (5) induces an operator $T$ on the space of sequences converging to $\mathbf{0}$, and the sequence $\{\mathbf{x}_k\}_{k \in \mathbb{N}}$ is the fixed point of $T$. This is so called the Lyapunov-Perron method (see supplementary material for some brief overview of the method). By Banach fixed point theorem (see supplementary material for the statement of the theorem), it can be proved that the sequence $\{\mathbf{x}_k\}_{k \in \mathbb{N}}$ (as the fixed point of $T$) exists and is unique. Furthermore, this implies that there is a unique $\mathbf{x}_0^-$ corresponding to $\mathbf{x}_0^+$, i.e. there exists a well defined function $\varphi : E^s \to E^u$ such that $\mathbf{x}_0^- = \varphi(\mathbf{x}_0^+)$.

## 4 Stable Manifold Theorem for Time Non-homogeneous Dynamical Systems

We start this section by showing the main technical result of this paper. This is a new stable manifold theorem that works for time non-homogeneous dynamical systems and is used to prove our main result (Theorem 1.1) for Gradient Descent, Mirror Descent, Proximal Point and Manifold Descent. The

proof of this theorem exploits the structure of the aforementioned first-order methods as dynamical systems.

**Theorem 4.1** (A new stable manifold theorem). *Let $H$ be a $d \times d$ real diagonal matrix with at least one negative eigenvalue, i.e. $H = diag\{\lambda_1, ..., \lambda_d\}$ with $\lambda_1 \geq \lambda_2 \geq ...\lambda_s > 0 \geq \lambda_{s+1} \geq ... \geq \lambda_d$ and assume $\lambda_d < 0$. Let $\eta(k, \mathbf{x})$ be a continuously differentiable function such that $\eta(k, \mathbf{0}) = \mathbf{0}$ and for each $\epsilon > 0$, there exists a neighborhood of $\mathbf{0}$ in which it holds*

$$\|\eta(k, \mathbf{x}) - \eta(k, \mathbf{y})\| \leq \alpha_k \epsilon \|\mathbf{x} - \mathbf{y}\|, \text{ for all naturals } k. \tag{6}$$

*Let $\{\alpha_k\}_{k \in \mathbb{N}}$ be a sequence of positive real numbers of order $\Omega\left(\frac{1}{k}\right)$ that converges to zero. We define the time non-homogeneous dynamical system*

$$\mathbf{x}_{k+1} = g(k, \mathbf{x}_k), \text{ where } g(k, \mathbf{x}) = (I - \alpha_k H)\mathbf{x} + \eta(k, \mathbf{x}). \tag{7}$$

*Suppose that $E = E^s \oplus E^u$, where $E^s$ is the span of the eigenvectors corresponding to negative eigenvalues of $H$, and $E^u$ is the span of the eigenvectors corresponding to nonnegative eigenvalues of $H$. Then there exists a neighborhood $U$ of $\mathbf{0}$ and a $C^1$-manifold $V(\mathbf{0})$ in $U$ that is tangent to $E^s$ at $\mathbf{0}$, such that for all $\mathbf{x}_0 \in V(\mathbf{0})$, $\lim_{k \to \infty} g(k, \mathbf{x}_k) = \mathbf{0}$. Moreover, $\bigcap_{k=0}^{\infty} \tilde{g}^{-1}(k, 0, U) \subset V(\mathbf{0})$.*

We can generalize Theorem 4.1 to the case where matrix $H$ is diagonalizable and for any fixed point $\mathbf{x}^*$ (instead of $\mathbf{0}$, using a shifting argument). The statement is given below.

**Corollary 4.2.** *Let $\{\alpha_k\}_{k \in \mathbb{N}}$ be a sequence of positive real numbers that converges to zero. Additionally, $\alpha_k \in \Omega\left(\frac{1}{k}\right)$. Let $g(k, \mathbf{x}) : \mathbb{R}^d \to \mathbb{R}^d$ be $C^1$ maps for all $k \in \mathbb{N}$ and*

$$\mathbf{x}_{k+1} = g(k, \mathbf{x}_k) \tag{8}$$

*be a time non-homogeneous dynamical system. Assume $\mathbf{x}^*$ is a fixed point, i.e. $g(k, \mathbf{x}^*) = \mathbf{x}^*$ for all $k \in \mathbb{N}$. Suppose the Taylor expansion of $g(k, \mathbf{x})$ at $\mathbf{x}^*$ in some neighborhood of $\mathbf{x}^*$,*

$$g(k, \mathbf{x}) = g(k, \mathbf{x}^*) + D_\mathbf{x} g(k, \mathbf{x}^*)(\mathbf{x} - \mathbf{x}^*) + \theta(k, \mathbf{x}), \text{ satisfies} \tag{9}$$

1. *$D_\mathbf{x} g(k, \mathbf{x}^*) = I - \alpha_k G$, $G$ real diagonalizable with at least one negative eigenvalue;*

2. *For each $\epsilon > 0$, there exists an open neighborhood centering at $\mathbf{x}^*$ of radius $\delta > 0$, denoted as $\mathbb{B}(\mathbf{x}^*, \delta)$, such that*

$$\|\theta(k, \mathbf{u}_1) - \theta(k, \mathbf{u}_2)\| \leq \alpha_k \epsilon \|\mathbf{u}_1 - \mathbf{u}_2\| \tag{10}$$

   *for all $k \in \mathbb{N}$ and all $\mathbf{u}_1, \mathbf{u}_2 \in \mathbb{B}(\mathbf{x}^*, \delta)$.*

*There exists a open neighborhood $U$ of $\mathbf{x}^*$ and a $C^1$-manifold $W(\mathbf{x}^*)$ in $U$, with codimension at least one, such that for $\mathbf{x}_0 \in W(\mathbf{x}^*)$, $\lim_{k \to \infty} g(k, \mathbf{x}^*) = \mathbf{x}^*$. Moreover, $\bigcap_{k=0}^{\infty} \tilde{g}^{-1}(k, 0, U) \subset W(\mathbf{x}^*)$.*

*Proof.* Since $G$ is diagonalizable, there exists invertible matrix $Q$ such that $G = Q^{-1}HQ$, hence $QGQ^{-1} = H$, where $H = diag\{\lambda_1, ..., \lambda_d\}$ (i.e., $H$ is a diagonal matrix with entries $\lambda_1, ..., \lambda_d$). Consider the map $\mathbf{z} = \varphi(\mathbf{x}) = Q(\mathbf{x} - \mathbf{x}^*)$. $\varphi$ induces a new dynamical system in terms of $\mathbf{z}$ as follows:

$$Q^{-1}\mathbf{z}_{k+1} = (I - \alpha_k G)Q^{-1}\mathbf{z}_k + \theta(k, Q^{-1}\mathbf{z}_k + \mathbf{x}^*).$$

Multiplying by $Q$ from the left on both sides, we have

$$\mathbf{z}_{k+1} = Q(I - \alpha_k G)Q^{-1}\mathbf{z}_k + Q\theta(k, Q^{-1}\mathbf{z}_k + \mathbf{x}^*) = (I - \alpha_k H)\mathbf{z}_k + \hat{\theta}(k, \mathbf{z}_k), \tag{11}$$

where $\hat{\theta}(k, \mathbf{z}_k) = Q\theta(k, Q^{-1}\mathbf{z}_k + \mathbf{x}^*)$. Denote $q(k, \mathbf{z}) = (I - \alpha_k H)\mathbf{z} + \hat{\theta}(k, \mathbf{z})$ the update rule given by equation (11). In order to apply Theorem 4.1, we next verify that $\hat{\theta}(k, \cdot)$ satisfies the condition (6) in Theorem 4.1 for all $k \in \mathbb{N}$. It is essentially to verify that given any $\epsilon > 0$, there exists a $\delta' > 0$, such that

$$\left\|\hat{\theta}(k, \mathbf{w}_1) - \hat{\theta}(k, \mathbf{w}_2)\right\| = \left\|Q\theta(k, Q^{-1}\mathbf{w}_1 + \mathbf{x}^*) - Q\theta(k, Q^{-1}\mathbf{w}_2 + \mathbf{x}^*)\right\| \leq \alpha_k \epsilon \|\mathbf{w}_1 - \mathbf{w}_2\| \tag{12}$$

for all $\mathbf{w}_1, \mathbf{w}_2 \in \mathbb{B}(0, \delta')$. Let's elaborate it. According to (10) of condition 2, for any given $\epsilon > 0$, and then $\frac{\epsilon}{\|Q\|\|Q^{-1}\|}$ is also a small positive number, there exists a $\delta > 0$ (w.r.t. $\frac{\epsilon}{\|Q\|\|Q^{-1}\|}$), such that

$$\|\theta(k, \mathbf{u}_1) - \theta(k, \mathbf{u}_2)\| \leq \alpha_k \frac{\epsilon}{\|Q\| \|Q^{-1}\|} \|\mathbf{u}_1 - \mathbf{u}_2\|$$

for all $\mathbf{u}_1, \mathbf{u}_2 \in \mathbb{B}(\mathbf{x}^*, \delta)$. Denote $V = Q(\mathbb{B}(\mathbf{x}^*, \delta) - \mathbf{x}^*)$, i.e.

$$V = \{\mathbf{w} \in \mathbb{R}^d : \mathbf{w} = Q(\mathbf{u} - \mathbf{x}^*) \text{ for some } \mathbf{u} \in \mathbb{B}(\mathbf{x}^*, \delta)\},$$

and it is easy to see that $\mathbf{0} \in V$. Since $Q(\mathbf{u} - \mathbf{x}^*)$ is a diffeomorphism (composition of a translation and a linear isomorphism) from the open ball $\mathbb{B}(\mathbf{x}^*, \delta)$ to $\mathbb{R}^d$, $V$ is an open neighborhood (not necessarily a ball) of $\mathbf{0}$. Therefore, there exists an open ball at $\mathbf{0}$ with radius $\delta'$, denoted as $\mathbb{B}(\mathbf{0}, \delta')$, such that $\mathbb{B}(\mathbf{0}, \delta') \subset V$. Next we show that $\mathbb{B}(\mathbf{0}, \delta')$ satisfies the inequality (12). By the definition of $V$, for any $\mathbf{w}_1, \mathbf{w}_2 \in \mathbb{B}(\mathbf{0}, \delta') \subset V$, there exist $\mathbf{u}_1, \mathbf{u}_2 \in \mathbb{B}(\mathbf{x}^*, \delta)$, such that

$$\mathbf{w}_1 = Q(\mathbf{u}_1 - \mathbf{x}^*), \quad \mathbf{w}_2 = Q(\mathbf{u}_1 - \mathbf{x}^*), \tag{13}$$

and the inverse transformation is given by $\mathbf{u}_1 = Q^{-1}\mathbf{w}_1 + \mathbf{x}^*$, $\mathbf{u}_2 = Q^{-1}\mathbf{w}_2 + \mathbf{x}^*$. Plugging to inequality (12), we have

$$
\begin{aligned}
\left\| \hat{\theta}(k, \mathbf{w}_1) - \hat{\theta}(k, \mathbf{w}_2) \right\| &= \left\| Q\theta(k, Q^{-1}\mathbf{w}_1 + \mathbf{x}^*) - Q\theta(k, Q^{-1}\mathbf{w}_2 + \mathbf{x}^*) \right\| \\
&= \left\| Q\theta(k, \mathbf{u}_1) - Q\theta(k, \mathbf{u}_2) \right\| \\
&\leq \|Q\| \, \|\theta(k, \mathbf{u}_1) - \theta(k, \mathbf{u}_2)\| \leq \|Q\| \, \alpha_k \frac{\epsilon \, \|\mathbf{u}_1 - \mathbf{u}_2\|}{\|Q\| \, \|Q^{-1}\|} \\
&= \|Q\| \, \alpha_k \frac{\epsilon}{\|Q\| \, \|Q^{-1}\|} \left\| (Q^{-1}\mathbf{w}_1 + \mathbf{x}^*) - (Q^{-1}\mathbf{w}_2 + \mathbf{x}^*) \right\| \\
&\leq \|Q\| \, \alpha_k \frac{\epsilon}{\|Q\| \, \|Q^{-1}\|} \left\| Q^{-1} \right\| \|\mathbf{w}_1 - \mathbf{w}_2\| = \alpha_k \epsilon \|\mathbf{w}_1 - \mathbf{w}_2\| .
\end{aligned}
$$

Thus the verification is complete. So as a consequence of Theorem 4.1, there exists a $C^1$-manifold $V(\mathbf{0})$ such that for all $\mathbf{z}_0 \in V(\mathbf{0})$, $\lim_{k \to \infty} \tilde{q}(k, 0, \mathbf{z}_0) = \mathbf{0}$. For the neighborhood $\varphi^{-1}(\mathbb{B}(\mathbf{0}, \delta'))$ of $\mathbf{x}^*$, denote $W(\mathbf{x}^*)$ the local stable set of dynamical system given by $g(k, \mathbf{x})$, i.e.,

$$W(\mathbf{x}^*) = \{\mathbf{x}_0 \in \varphi^{-1}(\mathbb{B}(\mathbf{0}, \delta)) : \lim_{k \to \infty} \tilde{g}(k, 0, \mathbf{x}_0) = \mathbf{x}^*\}.$$

We claim that $W(\mathbf{x}^*) \subset \varphi^{-1}(V(\mathbf{0}))$ and the proof is as follows:
Suppose $\mathbf{x}_0 \in W(\mathbf{x}^*)$, then the sequence $\{\mathbf{x}_k\}_{k \in \mathbb{N}}$ generated by $\mathbf{x}_{k+1} = g(k, \mathbf{x}_k)$ with initial condition $\mathbf{x}_0$ converges to $\mathbf{x}^*$. The map $\varphi$ induces a sequence $\{\mathbf{z}_k\}_{k \in \mathbb{N}}$, where $\mathbf{z}_0 = \varphi(\mathbf{x}_0)$ and

$$
\begin{aligned}
\mathbf{z}_{k+1} = \varphi(\mathbf{x}_{k+1}) &= \varphi\left(g(k, \mathbf{x}_k)\right) & \text{(14)} \\
&= Q\left(\mathbf{x}^* + (I - \alpha_k G)(\mathbf{x}_k - \mathbf{x}^*) + \theta(k, \mathbf{x}_k) - \mathbf{x}^*\right) & \text{(15)} \\
&\quad \text{(since } \mathbf{x}_k = \varphi^{-1}(\mathbf{z}_k) = Q^{-1}\mathbf{z}_k + \mathbf{x}^*) & \text{(16)} \\
&= Q(I - \alpha_k G)Q^{-1}\mathbf{z}_k + Q\theta(k, Q^{-1}\mathbf{z}_k + \mathbf{x}^*) = (I - \alpha_k H)\mathbf{z}_k + \hat{\theta}(k, \mathbf{z}_k). & \text{(17)}
\end{aligned}
$$

Since $\mathbf{z}_k = \varphi(\mathbf{x}_k)$, and $\mathbf{x}_k \to \mathbf{x}^*$, we have that $\mathbf{z}_k \to 0$. This implies sequence $\mathbf{z}_k$ generated by $\mathbf{z}_{k+1} = q(k, \mathbf{z}_k)$ with initial condition $\mathbf{z}_0$ converges to $\mathbf{0}$, meaning that $\mathbf{z}_0 = \varphi(\mathbf{x}_0) \in V(\mathbf{x}^*)$. Therefore $W(\mathbf{x}^*) \subset \varphi^{-1}(V(\mathbf{0}))$. Let $U = \varphi^{-1}(\mathbb{B}(\mathbf{0}, \delta))$ and the proof is complete. $\square$

We conclude this section by the following corollary which can be proved using standard arguments about separability of $\mathbb{R}^d$ (every open cover has a countable subcover). We denote $W^s(\mathcal{A}^*)$ the set of initial conditions so that the given dynamical system $g$ converges to a fixed point $\mathbf{x}^*$ such that matrix $D_{\mathbf{x}}g(k, \mathbf{x}^*)$ has an eigenvalue with absolute value greater than one for all $k$.

**Corollary 4.3.** *Let $g(k, \mathbf{x}) : \mathbb{R}^d \to \mathbb{R}^d$ be the mappings defined in Theorem 4.2. Then $W^s(\mathcal{A}^*)$ has Lebesgue measure zero.*

## 5  Applications

In this section, we apply Theorem 4.1 (or its corollary 4.2) to the four of the most commonly used first-order methods and we prove that each one of them avoids strict saddle points even with vanishing stepsize $\alpha_k$ of order $\Omega\left(\frac{1}{k}\right)$.

## 5.1 Gradient Descent

Let $f(\mathbf{x}) : \mathbb{R}^d \to \mathbb{R}$ be a real-valued $C^2$ function, and $g(k, \mathbf{x}) = \mathbf{x} - \alpha_k \nabla f(\mathbf{x})$ be the update rule of gradient descent, where $\{\alpha_k\}_{k \in \mathbb{N}}$ is a sequence of positive real numbers. Then

$$\mathbf{x}_{k+1} = \mathbf{x}_k - \alpha_k \nabla f(\mathbf{x}_k) \tag{18}$$

is a time non-homogeneous dynamical system.

**Theorem 5.1.** *Let $\mathbf{x}_{k+1} = g(k, \mathbf{x}_k)$ be the gradient descent algorithm defined by equation 18, and $\{\alpha_k\}_{k \in \mathbb{N}}$ be a sequence of positive real numbers of order $\Omega\left(\frac{1}{k}\right)$ that converges to zero. Then the stable set of strict saddle points has Lebesgue measure zero.*

*Proof.* We need to verify that the Taylor expansion of $g(k, \mathbf{x})$ at $\mathbf{x}^*$ satisfies the conditions of Corollary 4.2. Condition 1 is obvious since the Hessian $\nabla^2 f(\mathbf{x}^*)$ is diagonalizable and has at least one negative eigenvalue. It suffice to verify condition 2. Consider the Taylor expansion of $g(k, \mathbf{x})$ in a neighborhood $U$ of $\mathbf{x}^*$:

$$\begin{aligned}
g(k, \mathbf{x}) &= g(k, \mathbf{x}^*) + D_{\mathbf{x}} g(k, \mathbf{x}^*)(\mathbf{x} - \mathbf{x}^*) + \theta(k, \mathbf{x}) \\
&= \mathbf{x}^* + (I - \alpha_k \nabla^2 f(\mathbf{x}^*))(\mathbf{x} - \mathbf{x}^*) + \theta(k, \mathbf{x}) \\
&= \mathbf{x} - \alpha_k \nabla^2 f(\mathbf{x}^*)(\mathbf{x} - \mathbf{x}^*) + \theta(k, \mathbf{x}).
\end{aligned}$$

So we can write $\theta(k, \mathbf{x}) = g(k, \mathbf{x}) - \mathbf{x} + \alpha_k \nabla^2 f(\mathbf{x}^*)(\mathbf{x} - \mathbf{x}^*)$, and then the differential of $\theta(k, \mathbf{x})$ with respect to $\mathbf{x}$ is $D_{\mathbf{x}} \theta(k, \mathbf{x}) = D_{\mathbf{x}}(g(k, \mathbf{x}) - \mathbf{x}) + \alpha_k \nabla^2 f(\mathbf{x}^*) = -\alpha_k \nabla^2 f(\mathbf{x}) + \alpha_k \nabla^2 f(\mathbf{x}^*)$. From the Fundamental Theorem of Calculus and chain rule for multivariable functions, we have

$$\theta(k, \mathbf{x}) - \theta(k, \mathbf{y}) = \int_0^1 \frac{d}{dt} \theta(k, t\mathbf{x} + (1-t)\mathbf{y}) dt = \int_0^1 D_{\mathbf{z}} \theta(k, \mathbf{z})|_{\mathbf{z} = t\mathbf{x} + (1-t)\mathbf{y}} \cdot (\mathbf{x} - \mathbf{y}) dt.$$

By the assumption of $f$ being $C^2$, we have that $\nabla^2 f(\mathbf{x})$ is continuous everywhere. And then for any given $\epsilon > 0$, there exists a open ball $\mathbb{B}(\mathbf{x}^*)$ centering at $\mathbf{x}^*$, such that $\left\|\nabla^2 f(\mathbf{x}) - \nabla^2 f(\mathbf{x}^*)\right\|$ for all $\mathbf{x} \in \mathbb{B}(\mathbf{x}^*)$. And this implies that $\|D_{\mathbf{x}} \theta(k, \mathbf{x})\| \le \alpha_k \epsilon$ for all $\mathbf{x} \in \mathbb{B}(\mathbf{x}^*)$. Since $t\mathbf{x} + (1-t)\mathbf{y} \in \mathbb{B}(\mathbf{x}^*)$ if $\mathbf{x}, \mathbf{y} \in \mathbb{B}(\mathbf{x}^*)$, we have that $\left\|D_{\mathbf{z}} \theta(k, \mathbf{z})|_{\mathbf{z} = t\mathbf{x} + (1-t)\mathbf{y}}\right\| \le \alpha_k \epsilon$ for all $t \in [0, 1]$. By Cauchy-Schwarz inequality, we have

$$\begin{aligned}
\|\theta(k, \mathbf{x}) - \theta(k, \mathbf{y})\| &= \left\| \int_0^1 D_{\mathbf{z}} \theta(k, \mathbf{z})|_{\mathbf{z} = t\mathbf{x} + (1-t)\mathbf{y}} \cdot (\mathbf{x} - \mathbf{y}) dt \right\| \\
&\le \left( \int_0^1 \left\| D_{\mathbf{z}} \theta(k, \mathbf{z})|_{\mathbf{z} = t\mathbf{x} + (1-t)\mathbf{y}} \right\| dt \right) \cdot \|\mathbf{x} - \mathbf{y}\| = \alpha_k \epsilon \|\mathbf{x} - \mathbf{y}\|,
\end{aligned}$$

the verification completes. By Corollary 4.2 and Corollary 4.3, we conclude that the stable set of strict saddle points has Lebesgue measure zero. □

## 5.2 Mirror Descent

We consider mirror descent algorithm in this section. Let $\mathbf{D}$ be a convex open subset of $\mathbb{R}^d$, and $M = \mathbf{D} \cap \mathbf{A}$ for some affine space $\mathbf{A}$. Given a function $f : M \to \mathbb{R}$ and a mirror map $\Phi$, the mirror descent algorithm with vanishing step-size is defined as

$$\mathbf{x}_{k+1} = g(k, \mathbf{x}_k) := h(\nabla \Phi(\mathbf{x}_k) - \alpha_k \nabla f(\mathbf{x}_k)), \tag{19}$$

where $h(\mathbf{x}) = \operatorname{argmax}_{\mathbf{z} \in M} \langle \mathbf{z}, \mathbf{x} \rangle - \Phi(\mathbf{z})$.

**Definition 5.2** (Mirror Map). *We say that $\Phi$ is a mirror map if it satisfies the following properties.*

- *$\Phi : \mathbf{D} \to \mathbb{R}$ is $C^2$ and strictly convex.*

- *The gradient of $\Phi$ is surjective onto $\mathbb{R}^d$, that is $\nabla \Phi(D) = \mathbb{R}^d$.*

- *$\nabla_R \Phi$ diverges on the relative boundary of $M$, that is $\lim_{x \to \partial M} \|\nabla_R \Phi(x)\| = \infty$.*

**Theorem 5.3.** *Let $\mathbf{x}_{k+1} = g(k, \mathbf{x}_k)$ be the mirror descent algorithm defined by equation (19), and $\{\alpha_k\}_{k \in \mathbb{N}}$ be a sequence of positive real numbers of order $\Omega\left(\frac{1}{k}\right)$ that converges to zero. Then the stable set of strict saddle points has Lebesgue measure zero.*

## 5.3 Proximal Point

The proximal point algorithm is given by the iteration

$$\mathbf{x}_{k+1} = g(k, \mathbf{x}_k) := \arg\min_{\mathbf{z}} f(\mathbf{z}) + \frac{1}{2\alpha_k} \|\mathbf{x}_k - \mathbf{z}\|^2. \tag{20}$$

**Theorem 5.4.** *Let $\mathbf{x}_{k+1} = g(k, \mathbf{x}_k)$ be the proximal point algorithm defined by equation (20), and $\{\alpha_k\}_{k\in\mathbb{N}}$ be a sequence of positive real numbers of order $\Omega\left(\frac{1}{k}\right)$ that converges to zero. Then the stable set of strict saddle points has Lebesgue measure zero.*

## 5.4 Manifold Gradient Descent

Let $M$ be a submanifold of $\mathbb{R}^d$, and $T_{\mathbf{x}}M$ be the tangent space of $M$ at $\mathbf{x}$. $P_M$ and $P_{T_{\mathbf{x}}M}$ be the orthogonal projector onto $M$ and $T_{\mathbf{x}}M$ respectively. Assume that $f : M \to \mathbb{R}$ is extendable to neighborhood of $M$ and let $\bar{f}$ be a smooth extension of $f$ to $\mathbb{R}^d$. Suppose that the Riemannian metric on $M$ is induced by Euclidean metric of $\mathbb{R}^d$, then the Riemannian gradient $\nabla_R f(\mathbf{x})$ is the projection of the gradient of $f(\mathbf{x})$ on $\mathbb{R}^d$, i.e. $\nabla_R f(\mathbf{x}) = P_{T_{\mathbf{x}}M} \nabla f(\mathbf{x})$. Then the manifold gradient descent algorithm is:

$$\mathbf{x}_{k+1} = g(k, \mathbf{x}_k) := P_M(\mathbf{x}_k - \alpha_k P_{T_{\mathbf{x}_k}M} \nabla f(\mathbf{x}_k)). \tag{21}$$

**Theorem 5.5.** *Let $\mathbf{x}_{k+1} = g(k, \mathbf{x}_k)$ be the manifold gradient descent defined by equation (21), and $\{\alpha_k\}_{k\in\mathbb{N}}$ be a sequence of positive real numbers of order $\Omega\left(\frac{1}{k}\right)$ that converges to zero. Then the stable set of strict saddle points has measure zero.*

For the case when $M$ is not a submanifold of $\mathbb{R}^d$, the manifold gradient descent algorithm depends on the Riemannian metric $R$ defined intrinsically, i.e., $R$ is not induced by any ambient metric. Given $f : M \to \mathbb{R}$, the Riemannian gradient $\nabla_R f$ is defined to be the unique vector field such that $R(\nabla_R f, X) = \partial_X f$ for all vector field $X$ on $M$. In local coordinate systems $\mathbf{x}(p) = (x_1, ..., x_d)$, $p \in M$, the Riemannian gradient is written as $\nabla_R f(\mathbf{x}) = \left( R^{1j} \frac{\partial f}{\partial x_j}, ..., R^{dj} \frac{\partial f}{\partial x_j} \right) = \left( R^{ij} \right) \cdot \nabla f(\mathbf{x})$, where $\left( R^{ij} \right)$ is the inverse of the metric matrix at the point $\mathbf{x}$ and $R^{ij} \frac{\partial f}{\partial x_j} = \sum_j R^{ij} \frac{\partial f}{\partial x_j}$ as the Einstein convention. Then the update rule (in a local coordinate system) is

$$\mathbf{x}_{k+1} = g(k, \mathbf{x}_k) := \mathbf{x}_k - \alpha_k \left( R^{ij} \right) \cdot \nabla f(\mathbf{x}_k). \tag{22}$$

**Theorem 5.6.** *Let $\mathbf{x}_{k+1} = g(k, \mathbf{x}_k)$ be the manifold gradient descent defined by equation (22), and $\{\alpha_k\}_{k\in\mathbb{N}}$ be a sequence of positive real numbers of order $\Omega\left(\frac{1}{k}\right)$ that converges to zero. Then the stable set of strict saddle points has measure zero.*

# 6 Conclusion

In this paper, we generalize the results of [16] for the case of vanishing stepsizes. We showed that if the stepsize $\alpha_k$ converges to zero with order $\Omega\left(\frac{1}{k}\right)$, then gradient descent, mirror descent, proximal point and manifold descent still avoid strict saddles. We believe that this is an important result that was missing from the literature since in practice vanishing or adaptive stepsizes are commonly used. Our main result boils down to the proof of a Stable-manifold theorem 4.1 that works for time non-homogeneous dynamical systems and might be of independent interest. We leave as an open question the case of Block Coordinate Descent (as it also appears in [16]).

# 7 Acknowledgements

Ioannis Panageas acknowledges SRG ISTD 2018 136 and NRF fellowship for AI. Georgios Piliouras and Xiao Wang acknowledge MOE AcRF Tier 2 Grant 2016-T2-1-170, grant PIE-SGP-AI-2018-01 and NRF 2018 Fellowship NRF-NRFF2018-07. We thank Tony Roberts for pointers to the literature of stability of non-autonomous dynamical systems.

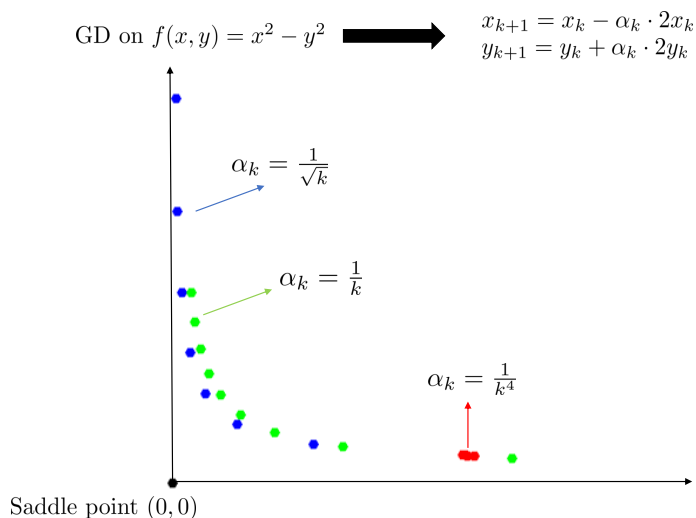

Figure 1: Steps of Gradient Descent for $x^2 - y^2$. $(0,0)$ is a strict saddle. Stepsizes $\frac{1}{\sqrt{k}}, \frac{1}{k}$ (blue, green) avoid $(0,0)$ (blue faster than green). Stepsize $\frac{1}{k^4}$ (red) converges to a non-critical point.

## Footnotes

[1]These are saddle points where the Hessian of the objective admits at least one direction of negative curvature. Such property has been shown to hold in a wide range of objective functions, see [11, 29, 28, 13, 12, 3] and references therein.

[2]This means that $g$ does not depend on time. In the dynamical systems/differential equations literature such systems are called "autonomous" whereas time-dependent systems are called "non-autonomous".

[3] $\{e_1, \ldots, e_d\}$ denote the classic orthogonal basis of $\mathbb{R}^d$.

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
