[Supplementary Material · supplementary.pdf]

# Supplementary Material for First-order methods almost always avoid saddle points: the case of vanishing step-sizes

**Ioannis Panageas**
SUTD
Singapore
ioannis@sutd.edu.sg

**Xiao Wang**
SUTD
Singapore
xiao_wang@sutd.edu.sg

**Georgios Piliouras**
SUTD
Singapore
georgios@sutd.edu.sg

## A  Statement of theorems used and for completion

**Theorem A.1** (Banach Fixed Point Theorem, 2.1 [2]). *Let $(X, d)$ be a complete metric space, then each contraction map $T : X \to X$ has unique fixed point.*

**Theorem A.2** (Center-Stable Manifold Theorem, III.7 [5]). *Let $x^*$ be a fixed point for the $C^r$ local diffeomorphism $g : \mathcal{X} \to \mathcal{X}$. Suppose that $E = E_s \oplus E_u$, where $E_s$ is the span of the eigenvectors corresponding to eigenvalues of magnitude less than or equal to one of $Dg(x^*)$, and $E_u$ is the span of the eigenvectors corresponding to eigenvalues of magnitude greater than one of $Dg(x^*)$[1]. Then there exists a $C^r$ embedded disk $W_{loc}^{cs}$ of dimension $dim(E^s)$ that is tangent to $E_s$ at $x^*$ called the local stable center manifold. Moreover, there exists a neighborhood $B$ of $x^*$, such that $g(W_{loc}^{cs}) \cap B \subset W_{loc}^{cs}$, and $\cap_{k=0}^{\infty} g^{-k}(B) \subset W_{loc}^{cs}$.*

## B  Lyapunov-Perron Method

The Lyapunov-Perron method has been developed by A.M. Lyapunov and O. Perron for the proof of the existence of stable and unstable manifolds of hyperbolic equilibrium points of ODEs. It uses the integral equation formulation of the differential equation and constructs the invariant manifold as a fixed point of an operator that is derived from this integral equation. The following case for time homogeneous ODEs can be found in Section 2.7, [4]. Let $F : \mathbb{R}^d \to \mathbb{R}^d$ be of $C^1$ with $F(\mathbf{0}) = \mathbf{0}$, consider the ODE

$$\frac{d\mathbf{x}}{dt} = F(\mathbf{x}),$$

whose linear approximation at $\mathbf{0}$ is

$$\frac{d\mathbf{x}}{dt} = A\mathbf{x} + \eta(\mathbf{x}).$$

By a change of coordinate system, $A$ is assumed to be decomposed to stable-unstable blocks respectively. Consider the operator $T$ defined as follows:

$$Tu(t, \mathbf{x}_0) = U(t)\mathbf{x}_0 + \int_0^t U(t-s)\eta(u(s, \mathbf{x}_0))ds - \int_t^{\infty} V(t-s)\eta(u(s, \mathbf{x}_0))ds$$

where $\mathbf{x}_0$ is the initial point, $U(t)$ and $V(t)$ are integral operators from the block decomposition of $A$. The stable manifold is the fixed point of $T$ following from the Banach fixed point theorem.

## C Missing Proofs

### C.1 Proof of Theorem 4.1

*Proof.* Denote $A(m,n) = (I - \alpha_m H) ... (I - \alpha_n H)$ for $m \geq n$, and $A(m,n) = I$ if $m < n$. Then the dynamical system can be written as

$$x_{k+1} = A(k,0)x_0 + \sum_{i=0}^{k} A(k,i+1)\eta(i,x_i). \tag{1}$$

Since $H$ is diagonal, the matrix $A(m,n)$ has the form of

$$\begin{pmatrix} B(m,n) & \\ & C(m,n) \end{pmatrix}$$

where $B_k$ and $C_k$ are diagonal as well and corresponding to *stable* and *unstable* subspaces of $I - \alpha_k H$ at 0. Using the same notation of denoting $A(m,n)$, we denote

$$B(m,n) = B_m \cdot ... \cdot B_n$$

and

$$C(m,n) = C_m \cdot ... \cdot C_n.$$

Let $v$ be a vector, we denote $v^+$ the stable component of $v$ and $v^-$ the unstable component of $v$. Then the solution (1) can be written in terms of stable and unstable components as

$$x_{k+1}^+ = B(k,0)x_0^+ + \sum_{i=0}^{k} B(k,i+1)\eta^+(i,x_i)$$

and

$$x_{k+1}^- = C(k,0)x_0^- + \sum_{i=0}^{k} C(k,i+1)\eta^-(i,x_i).$$

If $x_{k+1} \to 0$ as $k \to \infty$, then $x_{k+1}^- \to 0$ as $k \to \infty$. So we let $k \to \infty$, the following limit must holds:

$$\lim_k \left( C(k,0)x_0^- + \sum_{i=0}^{k} C(k,i+1)\eta^-(i,x_i) \right) = 0.$$

Then we can solve $x_0^-$ in limit:

$$x_0^- = \lim_k \left( C_0^{-1} \cdot ... \cdot C_k^{-1} x_{k+1}^{-1} - \left[ C_0^- \eta^-(0,x_0) + \cdots + C_0^{-1} \cdot ... \cdot C_k^{-1} \eta^-(k,x_k) \right] \right),$$

and then by taking limit as $k \to \infty$,

$$x_0^- = -\sum_{i=1}^{\infty} C(i-1,0)^{-1}\eta^-(i-1,x_{i-1}), \tag{2}$$

where $C(m,n)^{-1}$ denotes the inverse of $C(m,n)$.
So the initial condition $x_0$, if written as a column vector, has the form of

$$x_0 = \begin{pmatrix} x_0^+ \\ -\sum_{i=1}^{\infty} C(i-1,0)^{-1}\eta^-(i-1,x_{i-1}). \end{pmatrix}$$

Written as a column vecto, the solution of the dynamical system is of the form of

$$\begin{pmatrix} x_{k+1}^+ \\ x_{k+1}^- \end{pmatrix} = \begin{pmatrix} B(k,0)x_0^+ + \sum_{j=0}^{k} B(k,i+1)\eta^+(i,x_i) \\ C(k,0)x_0^- + \sum_{i=0}^{k} C(k,i+1)\eta^-(i,x_i). \end{pmatrix}$$

Plugging the equation (2) back to the above expression, we have

$$x_{k+1} = \left( \begin{array}{c} B\,(k,0)\,x_0^+ \\ -\sum_{i=0}^{k} C\,(k,i+1)\,\eta^-\,(i,x_i) - \sum_{i=0}^{\infty} C(k+1+i,k+1)^{-1}\eta^-\,(k+1+i,x_{k+1+i}) \end{array} \right) \tag{3}$$

$$+ \left( \begin{array}{c} \sum_{i=0}^{k} B\,(k,i+1)\,\eta^+\,(i,x_i) \\ \sum_{i=0}^{k} C\,(k,i+1)\,\eta^-\,(i,x_i) \end{array} \right) \tag{4}$$

$$= \left( \begin{array}{c} B\,(k,0)\,x_0^+ + \sum_{i=0}^{k} B\,(k,i+1)\,\eta^+\,(i,x_i) \\ -\sum_{i=0}^{\infty} C(k+1+i,k+1)^{-1}\eta^-\,(k+1+i,x_{k+1+i}). \end{array} \right) \tag{5}$$

Denote $\mathbb{B}(\delta) \subset \mathbb{R}^d$ the ball around 0 with Euclidean radius $\delta$. Denote

$$\ell_0(\mathbb{B}(\delta)) = \{\{u_n\}_{n\in\mathbb{N}} \subset \mathbb{B}(\delta) : \lim_{n\to\infty} u_n = 0\}$$

the metric space of sequences whose entries are in $\mathbb{B}(\delta)$, with metric defined as

$$d(u,v) := \sup_{n\geq 0}\{\|u_n - v_n\|\}. \tag{6}$$

for any $u = \{u_n\}_{n\in\mathbb{N}}$ and $v = \{v_n\}_{n\in\mathbb{N}}$ in the ball $\mathbb{B}(\delta)$. Then $\ell_0(\mathbb{B}(\delta))$ is a complete metric space
Reason is as follows:
Let $u_1 = \{u_{1j}\}_{j\in\mathbb{N}}$, $u_2 = \{u_{2j}\}_{j\in\mathbb{N}}$,..., $u_i = \{u_{ij}\}_{j\in\mathbb{N}}$,... be a sequence of sequences in $\ell_0(\mathbb{B}(\delta))$. Suppose $\{u_i\}_{i\in\mathbb{N}}$ is Cauchy with respect to the metric defined by 6,i.e. given any $\epsilon > 0$, there exists integer $L > 0$, such that

$$d(u_n, u_m) = \sup_{j\geq 0}\{\|u_{nj} - u_{mj}\|\} < \epsilon$$

for all $n, m > L$. This means that for each $j$, there exists a point $u_{*j} \in \mathbb{B}(\delta)$ such that $\lim_{i\to\infty} u_{ij} = u_{*j}$. And then we denote the limit sequence as $u_* = \{u_{*j}\}_{j\in\mathbb{N}}$. Furthermore, letting $m \to \infty$, we have that

$$\|u_{nj} - u_{*j}\| < \epsilon$$

for all $n > L$. Fixing $n$ and letting $j \to \infty$, we have $u_{*j} \to 0$ since $u_{nj} \to 0$. And this shows that $u_* \in \ell_0(\mathbb{B}(\delta))$.

Define the operator $T$ for each sequence $x = \{x_n\}_{n\in\mathbb{N}} \subset \mathbb{R}^d$ to be

$$(Tx)_{k+1} = \left( \begin{array}{c} B\,(k,0)\,x_0^+ + \sum_{i=0}^{k} B\,(k,i+1)\,\eta^+\,(i,x_i) \\ -\sum_{i=0}^{\infty} C(k+1+i,k+1)^{-1}\eta^-\,(k+1+i,x_{k+1+i}). \end{array} \right) \tag{7}$$

for $k \geq 0$ and $(Tx)_0 = x_0$.
Next we prove that $T$ is a contraction map when choosing sequence in a small enough neighborhood around 0.
Take $\mathbb{B}(\delta)$ a small enough neighborhood around 0 such that the Lipschitz condition is satisfied. Let $u = \{u_n\}_{n\in\mathbb{N}} \subset \mathbb{B}(\delta)$ and $v = \{v_n\}_{n\in\mathbb{N}} \subset \mathbb{B}(\delta)$. Then we have

$$(Tu - Tv)_{k+1} = (Tu)_{k+1} - (Tv)_{k+1} \tag{8}$$

$$= \left( \begin{array}{c} B\,(k,0)\,u_0^+ + \sum_{i=0}^{k} B\,(k,i+1)\,\eta^+\,(i,u_i) \\ -\sum_{i=0}^{\infty} C(k+1+i,k+1)^{-1}\eta^-\,(k+1+i,u_{k+1+i}) \end{array} \right) \tag{9}$$

$$- \left( \begin{array}{c} B\,(k,0)\,v_0^+ + \sum_{i=0}^{k} B\,(k,i+1)\,\eta^+\,(i,v_i) \\ -\sum_{i=0}^{\infty} C(k+1+i,k+1)^{-1}\eta^-\,(k+1+i,v_{k+1+i}) \end{array} \right) \tag{10}$$

$$= \left( \begin{array}{c} B\,(k,0)\,(u_0^+ - v_0^+) + \sum_{i=0}^{k} B\,(k,i+1)\,(\eta^+(i,u_i) - \eta^+\,(i,v_i)) \\ -\sum_{i=0}^{\infty} C(k+1+i,k+1)^{-1}(\eta^-(k+1+i,u_{k+1+i}) - \eta^-\,(k+1+i,v_{k+1+i})). \end{array} \right) \tag{11}$$

Use spectrum norm $\|\cdot\|$ for matrices, we have

$$|(Tu - Tv)_{k+1}| \leq \|B(k,0)\| \left|u_0^+ - v_0^+\right| + \sum_{i=0}^{k} \|B(k,i+1)\| \left\|\eta^+(i,u_i) - \eta^+(i,v_i)\right\| \tag{12}$$

$$+ \sum_{i=0}^{\infty} \left\|C(k+1+i,k+1)^{-1}\right\| \left\|\eta^-(k+1+i,u_{k+1+i}) - \eta^-(k+1+i,v_{k+1+i})\right\| \tag{13}$$

$$\text{(by Lipschitz assumption (6))} \tag{14}$$

$$\leq \|B(k,0)\| \left\|u_0^+ - v_0^+\right\| + \sum_{i=0}^{k} \|B(k,i+1)\| \, \alpha_i \epsilon \, \|u_i - v_i\| \tag{15}$$

$$+ \sum_{i=0}^{\infty} \left\|C(k+1+i,k+1)^{-1}\right\| \alpha_{k+1+i}\epsilon \, \|u_{k+1+i} - v_{k+1+i}\| \tag{16}$$

$$\leq \|B(k,0)\| \, d(u,v) + \sum_{i=0}^{k} \|B(k,i+1)\| \, \alpha_i \epsilon d(u,v) \tag{17}$$

$$+ \sum_{i=0}^{\infty} \left\|C(k+1+i,k+1)^{-1}\right\| \alpha_{k+1+i}\epsilon d(u,v) \tag{18}$$

$$= \|B(k,0)\| \, d(u,v) + \sum_{i=0}^{k} \alpha_i \epsilon \, \|B(k,i+1)\| \, d(u,v) \tag{19}$$

$$+ \sum_{i=0}^{\infty} \alpha_{k+1+i}\epsilon \left\|C(k+1+i,k+1)^{-1}\right\| d(u,v) \tag{20}$$

$$= \|B(k,0)\| \, d(u,v) + \epsilon d(u,v) \left( \sum_{i=0}^{k} \alpha_i \, \|B(k,i+1)\| \right) \tag{21}$$

$$+ \epsilon d(u,v) \left( \sum_{i=0}^{\infty} \alpha_{k+1+i} \left\|C(k+1+i,k+1)^{-1}\right\| \right) \tag{22}$$

Next we proceed to prove that

$$\|B(k,0)\| + \epsilon \left( \sum_{i=0}^{k} \alpha_i \, \|B(k,i+1)\| \right) + \epsilon \left( \sum_{i=0}^{\infty} \alpha_{k+1+i} \left\|C(k+1+i,k+1)^{-1}\right\| \right)$$

can be taken less than 1 so that $T$ is a contraction map on $\ell_0(\mathbb{B}(\delta))$.

**Lemma C.1.**

$$R_k = \sum_{i=0}^{\infty} \alpha_{k+1+i} \left\|C(k+1+i,k+1)^{-1}\right\|$$

*is a convergent series for each $k \in \mathbb{N}^+$. Moreover, there exists a constant $K_2 > 0$ such that $R_k \leq K_2$ for all $k \in \mathbb{N}^+$.*

*Proof.* Denote $\lambda$ the least negative eigenvalue, then the spectrum norm of $C(k+1+i,k+1)^{-1}$ is

$$\left\|C(k+1+i,k+1)^{-1}\right\| = \prod_{j=k+1}^{k+1+i} (1 - \alpha_j \lambda)^{-1}.$$

Since the sequence $\alpha_i$ is chosen to be small, we have

$$R_k = \sum_{i=0}^{\infty} \alpha_{k+1+i} \left\| C(k+1+i, k+1)^{-1} \right\| \tag{23}$$

$$\leq \alpha_k \sum_{i=0}^{\infty} \left\| C(k+1+i, k+1)^{-1} \right\| \tag{24}$$

$$= \alpha_k \sum_{i=0}^{\infty} \prod_{j=k+1}^{k+1+i} (1 - \alpha_j \lambda)^{-1}. \tag{25}$$

Using the inequality $1 + x \leq e^x$, we have

$$(1 - \alpha_j \lambda)^{-1} = \frac{1}{1 - \alpha_j \lambda} = 1 + \frac{\alpha_j \lambda}{1 - \alpha_j \lambda} \leq \exp\left( \frac{\alpha_j \lambda}{1 - \alpha_j \lambda} \right)$$

and

$$\prod_{j=k+1}^{k+1+i} (1 - \alpha_j \lambda)^{-1} \leq \exp\left( \sum_{j=k+1}^{k+1+i} \frac{\alpha_j \lambda}{1 - \alpha_j \lambda} \right),$$

and thus

$$R_k \leq \alpha_k \sum_{i=0}^{\infty} \exp\left( \sum_{j=k+1}^{k+1+i} \frac{\alpha_j \lambda}{1 - \alpha_j \lambda} \right). \tag{26}$$

Since by assumption, $\alpha_j \in \Omega\left( \frac{1}{j^p} \right)$ and $\lambda < 0$, so $1 - \alpha_j \lambda$ is positive and bounded, i.e. $1 < 1 - \alpha_j \lambda < c$. And then the following inequalities hold:

$$\frac{\alpha_j}{1 - \alpha_j \lambda} \geq \frac{1}{1 - \alpha_j \lambda} \cdot \frac{1}{j^p} \geq \frac{1}{c} \cdot \frac{1}{j^p}.$$

Multiplying by the negative number $\lambda$, we have

$$\frac{\alpha_j \lambda}{1 - \alpha_j \lambda} \leq \frac{\lambda}{c} \cdot \frac{1}{j^p}.$$

Combining with the inequality 26, we obtain

$$R_k \leq \alpha_k \sum_{i=0}^{\infty} \exp\left( \sum_{j=k+1}^{k+1+i} \frac{\alpha_j \lambda}{1 - \alpha_j \lambda} \right) \leq \alpha_k \sum_{i=0}^{\infty} \exp\left( \frac{\lambda}{c} \sum_{j=k+1}^{k+1+i} \frac{1}{j^p} \right).$$

By definition of definite integral, we notice that

$$\sum_{j=k+1}^{k+1+i} \frac{1}{j^p} > \int_{k+1}^{k+2+i} \frac{1}{t^p} dt \tag{27}$$

$$= \frac{1}{1-p}(k+2+i)^{1-p} - \frac{1}{1-p}(k+1)^{1-p}. \tag{28}$$

Since $\lambda < 0$,

$$\frac{\lambda}{c} \sum_{j=k+1}^{k+1+i} \frac{1}{j^p} < \frac{\lambda}{c(1-p)}(k+2+i)^{1-p} - \frac{\lambda}{c(1-p)}(k+1)^{1-p}$$

so we have

$$\exp\left( \frac{\lambda}{c} \sum_{j=k+1}^{k+1+i} \frac{1}{j^p} \right) < \exp\left( \frac{\lambda}{c(1-p)}(k+2+i)^{1-p} - \frac{\lambda}{c(1-p)}(k+1)^{1-p} \right) \tag{29}$$

$$= \exp\left( \frac{\lambda}{c(1-p)}(k+2+i)^{1-p} \right) \cdot \exp\left( -\frac{\lambda}{c(1-p)}(k+1)^{1-p} \right). \tag{30}$$

So for each fixed $k$, we have that

$$R_k \le \alpha_k \sum_{i=0}^{\infty} \exp\left(\frac{\lambda}{c} \sum_{j=k+1}^{k+1+i} \frac{1}{j^p}\right) \tag{31}$$

$$< \alpha_k \exp\left(-\frac{\lambda}{c(1-p)}(k+1)^{1-p}\right) \cdot \sum_{i=0}^{\infty} \exp\left(\frac{\lambda}{c(1-p)}(k+2+i)^{1-p}\right). \tag{32}$$

The series

$$\sum_{i=0}^{\infty} \exp\left(\frac{\lambda}{c(1-p)}(k+2+i)^{1-p}\right) \tag{33}$$

has the same convergence as the integral

$$\int_k^{\infty} \exp\left(-t^{1-p}\right) dt.$$

Notice that

$$\int_k^{\infty} \exp\left(-t^{1-p}\right) dt = \int_{k^{1-p}}^{\infty} \exp(-u) \frac{1}{1-p} u^{\frac{1}{1-p}-1} du \tag{34}$$

$$= \frac{1}{1-p} \int_{k^{1-p}}^{\infty} \exp(-u) u^{\frac{1}{1-p}-1} du \tag{35}$$

$$= \frac{1}{1-p} \Gamma\left(\frac{1}{1-p}, k^{1-p}\right), \text{(the incomplete Gamma function)} \tag{36}$$

which implies that $\int_k^{\infty} \exp(-t^{1-p}) dt$ converges, so does the series 33.

Since the incomplete Gamma function $\Gamma(s, x)$ has the property

$$\frac{\Gamma(s, x)}{x^{s-1} e^{-x}} \to 1 \quad \text{as} \quad x \to \infty,$$

let $s = \frac{1}{1-p}$ and $x = k^{1-p}$ so that $x^{s-1} = (k^{1-p})^{\frac{1}{1-p}-1} = k^p$, we have that

$$\frac{1}{k^p} e^{k^{1-p}} \Gamma\left(\frac{1}{1-p}, k^{1-p}\right) = \frac{\Gamma(\frac{1}{1-p}, k^{1-p})}{k^p e^{-k^{1-p}}} \to 1$$

as $k \to \infty$. This implies that $R_k$ is bounded as $k \to \infty$. $\qquad\square$

**Lemma C.2.** *The sequence*

$$S_k = \sum_{i=0}^{k} \alpha_i \|B(k, i+1)\|$$

*is uniformly bounded for all $k \in \mathbb{N}$, i.e. there exists positive number $K_1$ such that $S_k \le K_1$*

*Proof.* Since $B(k, i+1)$ is diagonal, denote $\lambda$ the least positive eigenvalue of $H$, by definition of $B(k, i+1)$, we have that

$$\|B(k, i+1)\| = (1 - \alpha_k \lambda) \cdots (1 - \alpha_{i+1} \lambda).$$

Then

$$S_k = \alpha_0 (1 - \alpha_k \lambda) \cdots (1 - \alpha_1 \lambda) + \ldots + \alpha_k.$$

Notice that

$$S_{k+1} = (1 - \alpha_{k+1} \lambda) S_k + \alpha_{k+1}.$$

Consider the difference between $S_{k+1}$ and $S_k$, we have

$$S_{k+1} - S_k = (1 - \alpha_{k+1} \lambda) S_k + \alpha_{k+1} - S_k \tag{37}$$

$$= S_k - \alpha_{k+1} \lambda S_k + \alpha_{k+1} - S_k \tag{38}$$

$$= \alpha_{k+1}(1 - \lambda S_k). \tag{39}$$

We observe the following facts:

1. If $S_k = \frac{1}{\lambda}$, then $S_k = S_{k+1} \equiv \frac{1}{\lambda}$.

2. If $S_k > \frac{1}{\lambda}$, or equivalently $1 - \lambda S_k < 0$, then $S_{k+1} - S_k < 0$, and $S_k$ decreases until $S_{k_1} < \frac{1}{\lambda}$ for some $k_1 \in \mathbb{N}$.

3. If $S_k < \frac{1}{\lambda}$, or equivalently $1 - \lambda S_k > 0$, then $S_{k+1} - S_k > 0$, and $S_k$ increases until $S_k > \frac{1}{\lambda}$.

So $S_k$ decreases or increases to $\frac{1}{\lambda}$ (meaning that $S_k$ is bounded), or $S_k$ oscillates around $\frac{1}{\lambda}$. Suppose that $S_k < \frac{1}{\lambda}$ and $S_{k+1} > \frac{1}{\lambda}$, we have that

$$S_{k+1} \leq S_k + \alpha_{k+1} \leq \frac{1}{\lambda} + \frac{1}{\lambda} = \frac{2}{\lambda}$$

when $k$ is large so that $\alpha_k < \frac{1}{\lambda}$. Then in conclusion, $S_k$ is bounded, and the proof completes. $\square$

Next result shows that $T$ maps a sequence converging to 0 to another sequence converging to 0. And this is a prerequisite for $T$ to be a well defined map on the complete metric space $\ell_0(\mathbb{B}(\delta))$ to itself.

**Lemma C.3.** *Suppose $x = \{x_k\}_{k \in \mathbb{N}}$ and $\lim_{k \to \infty} x_k = 0$. Then $\lim_{k \to \infty} (Tx)_{k+1} = 0$*

*Proof.* Denote $(Tx)_{k+1}^+$ and $(Tx)_{k+1}^-$ the stable and unstable component of $(Tx)_{k+1}$ respectively. We prove $\lim_{k \to \infty} (Tx)_{k+1}^+ = 0$ and $\lim_{k \to \infty} (Tx)_{k+1}^- = 0$ separately.

1. $\lim_{k \to \infty} (Tx)_{k+1}^+ = 0$:
According to the definition of $T$ in 7,

$$(Tx)_{k+1}^+ = B(k, 0)x_0^+ + \sum_{i=0}^{k} B(k, i+1)\eta^+(i, x_i).$$

Since $\|B(k, 0)\| \to 0$ as $k \to \infty$, it is enough to show that

$$\sum_{i=0}^{k} B(k, i+1)\eta^+(i, x_i) \to 0$$

as $k \to \infty$. From the Lipschitz condition on $\eta$, we have that

$$\left\| \sum_{i=0}^{k} B(k, i+1)\eta^+(i, x_i(x_0)) \right\| \leq \epsilon \sum_{i=0}^{k} \|B(k, i+1)\| \cdot \alpha_i \|x_i\| \tag{40}$$

$$= \epsilon \left( (1 - \alpha_k \lambda) \cdots (1 - \alpha_1 \lambda)\alpha_0 \|x_0\| + \cdots (1 - \alpha_k \lambda)\alpha_{k-1} \|x_{k-1}\| + \alpha_k \|x_k\| \right). \tag{41}$$

Denote the sum above as

$$S_k = (1 - \alpha_k \lambda) \cdots (1 - \alpha_1 \lambda)\alpha_0 \|x_0\| + \cdots (1 - \alpha_k \lambda)\alpha_{k-1} \|x_{k-1}\| + \alpha_k \|x_k\| .$$

Notice that

$$S_{k+1} = (1 - \alpha_{k+1}\lambda)S_k + \alpha_{k+1} \|x_{k+1}\| ,$$

and then

$$S_{k+1} - S_k = \alpha_{k+1}(\|x_{k+1}\| - \lambda S_k).$$

From the proof of Lemma C.2, we know that $S_k$ is bounded, and thus $|S_{k+1} - S_k| \to 0$ as $k \to \infty$. Similar to proof of C.2, we have the following observation:

1. If $S_{k+1} - S_k > 0$, then $|x_{k+1}| - \lambda S_k > 0$, or $S_k < \frac{|x_k|}{\lambda}$;

2. If $S_{k+1} - S_k < 0$, then $|x_{k+1}| - \lambda S_k < 0$, or $S_k > \frac{|x_k|}{\lambda}$;

3. If $S_{k+1} - S_k = 0$, then $S_k = \text{constant}$.

So the sequence $S_k$ is either

1. decreasing but $S_k > \frac{\|x_k\|}{\lambda}$,

2. oscillating around $\frac{\|x_k\|}{\lambda}$.

If $S_k$ is of case 1, then $\lim_k S_k$ exists. Suppose that this limit is positive, but since we have $\sum \alpha_k = \infty$ and

$$S_{k+1} = S_k + \alpha_{k+1}(\|x_{k+1}\| - \lambda S_k)$$

implying that $S_k \to \infty$. So we conclude that $\lim_k S_k = 0$, contradicting to the fact that $\lim_k S_k$ exists. So the $\lim_k S_k$ must be 0 if $S_k$ is of case 1.

If $S_k$ is of case 2, then immediately $\liminf S_k = 0$. Suppose that $\limsup S_k > 0$. Since $S_k$ decreases whenever $S_k > \frac{\|x_k\|}{\lambda}$ and $S_k$ increases whenever $S_k < \frac{\|x_k\|}{\lambda}$, we can find a subsequence $S_{k_m}$, with $S_{k_m-1} < S_{k_m}$, converging to $\limsup S_k$ as $m \to \infty$. But this is impossible since $S_{k_m-1} < \frac{\|x_k\|}{\lambda}$ and then $S_{k_m-1} \to 0$ as $m \to \infty$, which means $\lim_m |S_{k_m-1} - S_{k_m}|$ is positive, contradicting to the fact that $\lim_k |S_k - S_{k+1}| = 0$. And thus, we have $\limsup_k S_k = 0$, meaning that $\lim_k S_k = 0$.

So we conclude that either in case 1 or 2, the limit $\lim_k S_k = 0$, which completes the proof of part 1.

2. $\lim_{k \to \infty}(Tx)_{k+1}^- = 0$:
According to the equation 7,

$$(Tx)_{k+1}^- = -\sum_{i=0}^{\infty} C(k+1+i, k+1)^{-1} \eta^-(k+1+i, x_{k+1+i}).$$

And from the Lipschitz condition of on $\eta$, we have that

$$\left\|(Tx)_{k+1}^-\right\| \leq \sum_{i=0}^{\infty} \left\|C(k+1+i, k+1)^{-1}\right\| \left\|\eta^-(k+1+i, x_{k+1+i})\right\| \tag{42}$$

$$\leq \sum_{i=0}^{\infty} \left\|C(k+1+i, k+1)^{-1}\right\| \left\|\eta(k+1+i, x_{k+1+i})\right\| \tag{43}$$

$$\leq \sum_{i=0}^{\infty} \left\|C(k+1+i, k+1)^{-1}\right\| \epsilon \alpha_{k+1+i} \left\|x_{k+1+i}\right\| \tag{44}$$

$$\leq \sum_{i=0}^{\infty} \left\|C(k+1+i, k+1)^{-1}\right\| \epsilon \alpha_{k+1+i} \sup_{n>k} \|x_n\| \tag{45}$$

$$\leq \sup_{n>k} |x_n| \cdot K_2 \quad \text{(Lemma C.1)} \tag{46}$$

Since $\{x_n\}$ converges to 0 as $n \to \infty$, $\sup_{n>k} |x_n| \to 0$ as $k \to \infty$. And this completes the proof of part 2. $\qquad \square$

**Lemma C.4.** *There exists a real number $\delta > 0$ such that the operator $T$ given by equation 7*

$$T : \ell_0(\mathbb{B}(\delta)) \to \ell_0(\mathbb{B}(\delta))$$

*is a contraction map.*

*Proof.* From Lemma C.1 and Lemma C.2, we know that in equation (22),

$$\sum_{i=0}^{k} \alpha_i \|B(k, i+1)\| \leq K_1 \tag{47}$$

and

$$\sum_{i=0}^{\infty} \alpha_{k+1+i} \left\|C(k+1+i, k+1)^{-1}\right\| \leq K_2. \tag{48}$$

Since $B(k,0)$ is on the stable subspace and whose norm is calculated by

$$\|B(k,0)\| = \prod_{i=0}^{k}(1-\alpha_i\lambda)$$

where $\lambda > 0$, we have

$$\|B(k,0)\| \leq \|B(0,0)\| = 1 - \alpha_0\lambda < 1.$$

Then we can choose small positive $\epsilon$ so that

$$\epsilon < \frac{\alpha_0\lambda}{K_1 + K_2}.$$

Define the constant $K$ to be

$$K := 1 - \alpha_0\lambda + \epsilon(K_1 + K_2), \tag{49}$$

and by the choice of $\epsilon$, we know that $K < 1$. Let $\delta > 0$ be the radius corresponding to $\epsilon$ so that the Lipschitz condition is satisfied.
Combining 22, 47 and 48, we have that

$$\|(Tu - T_v)_{k+1}\| \leq (\|B(k,0)\| + \epsilon(K_1 + K_2))\, d(u,v) \leq Kd(u,v).$$

Since above $k$ is taken arbitrarily, we conclude that

$$\|Tu - Tv\| \leq Kd(u,v).$$

So $T$ is a contraction map. $\qquad\qquad\qquad\qquad\qquad\qquad\qquad\qquad\qquad\qquad\qquad\qquad\qquad\square$

And since $\ell_0(\mathbb{B}(\delta))$ is a complete metric space, according to Banach fixed point theorem, there exists a unique sequence, denoted as $x = \{x_n\}_{n\in\mathbb{N}}$, such that

$$Tx = x$$

with initial condition satisfying

$$(x_0^+, x_0^-) = (x_0^+, -\sum_{i=0}^{\infty} C(k+1+i, k+1)^{-1}\eta^-(k+1+i, x_{k+1+i})). \tag{50}$$

If we consider the sequence $x$ as a sequence of functions with the initial condition as the variable, the general term $x_n$ is written as $x_n(x_0)$, then the equation (50) is written as

$$(x_0^+, x_0^-) = \left(x_0^+, -\sum_{i=0}^{\infty} C(k+1+i, k+1)^{-1}\eta^-\left(k+1+i, x_{k+1+i}(x_0^+, x_0^-)\right)\right).$$

This means that if the some initial condition $x_0$ goes to 0 through the discrete time process $\{x_n(x_0)\}$, its stable and unstable component must satisfy following relation:

$$x_0^- = -\sum_{i=0}^{\infty} C(k+1+i, k+1)^{-1}\eta^-\left(k+1+i, x_{k+1+i}(x_0^+, x_0^-)\right).$$

Denote

$$\Phi(x_0^+, x_0^-) = -\sum_{i=0}^{\infty} C(k+1+i, k+1)^{-1}\eta^-\left(k+1+i, x_{k+1+i}(x_0^+, x_0^-)\right)$$

and the equation $x_0^- = \Phi(x_0^+, x_0^-)$ defines an implicit function $x_0^- = \varphi(x_0^+)$ by the uniqueness of Banach fixed point. Next we will show that $\varphi$ is differentiable with respect to $x_0^+$. Since it is enough to show the function $\Phi(x_0^+, x_0^-)$ is differentiable with respect to $x_0^+$. And it is enough to show that each $x_n(x_0)$, if considered as a function of initial condition $x_0$, is differentiable with respect to $x_0^+$.

**Lemma C.5.** *The solution $x_n(x_0^+, x_0^-)$ is of $C^1$ with respect to $x_0^+$ provided $\eta(n,x)$ is of $C^1$.*

*Proof.* It is equivalent to show that $\frac{\partial x_n}{\partial x_{0,j}}$, $j = 1, .., d$, where $d$ is the dimension of stable vector space, exist and are continuous for small $|x_0|$.

Let $P^+$ and $P^-$ be the projection operators to the stable and unstable subspaces respectively, then the solution (with initial condition $x_0$) of the dynamical system can be written as

$$x_{k+1}(x_0) = A(k,0)P^+ x_0 + \sum_{i=0}^{k} A(k, i+1)P^+ \eta(i, x_i(x_0)) \tag{51}$$

$$-\sum_{i=0}^{\infty} A(k+1+i, k+1)^{-1} P^- \eta(k+1+i, x_{k+1+i}(x_0)). \tag{52}$$

Let $h$ be a scalar and $e_j$ be the $j$th standard basis. Denote

$$q(n, x_0, h) = \frac{x_n(x_0 + he_j) - x_n(x_0)}{h}.$$

Notice the following identity holds:

$$\frac{\eta(n, x_n(x_0 + he_j)) - \eta(n, x_n(x_0))}{h} = \frac{\eta(n, x_n(x_0 + he_j)) - \eta(n, x_n(x_0))}{h} \tag{53}$$

$$+ D\eta(n, x_n(x_0))q(n, x_0, h) \tag{54}$$

$$- D\eta(n, x_n(x_0))q(n, x_0, h). \tag{55}$$

Plugging above identity to 51, we can compute the difference quotient $q(k+1, x_0, h) = \frac{x_{k+1}(x_0 + he_j) - x_{k+1}(x_0)}{h}$

$$q(k+1, x_0, h) = A(k,0)P^+ \left( \frac{(x_0 + he_j) - x_0}{h} \right) \tag{56}$$

$$+ \sum_{i=0}^{k} A(k, i+1)P^+ \left( \frac{\eta(i, x_i(x_0 + he_j)) - \eta(i, x_i(x_0))}{h} \right) \tag{57}$$

$$- \sum_{i=0}^{\infty} A(k+1+i, k+1)^{-1} P^- \left( \frac{\eta(k+1+i, x_{k+1+i}(x_0 + he_j)) - \eta(k+1+i, x_{k+1+i}(x_0))}{h} \right) \tag{58}$$

$$= A(k,0)P^+ e_j + \sum_{i=0}^{k} A(k, i+1)P^+ \left( D\eta(i, x_i(x_0))q(i, x_0, h) + \Delta_i \right) \tag{59}$$

$$- \sum_{i=0}^{\infty} A(k+1+i, k+1)^{-1} P^- \left( D\eta(k+1+i, x_{k+1+i}(x_0))q(k+1+i, x_0, h) + \Delta_{k+1+i} \right), \tag{60}$$

where

$$\Delta_n = \frac{\eta(n, x_n(x_0 + he_j)) - \eta(n, x_n(x_0))}{h} - D\eta(n, x_n(x_0))q(n, x_0, h).$$

Since for the solution $x(n, x_0)$, $x(n, x_0) \to 0$ as $n \to \infty$, and $\|\eta(n, x) - \eta(n, \bar{x})\| \le \epsilon \|x - \bar{x}\|$, we have that

$$\|D\eta(n, x_n(x_0))\| \le \epsilon d.$$

Given $\delta > 0$, $|h|$ can be chosen small that by the mean value theorem and the continuity of $D\eta$, we have

$$\|\Delta_n\| \le \frac{1}{h} \|D\eta(n, x')\| \cdot \|x_n(x_0 + he_j) - x_n(x_0)\| + \|D\eta(n, x_n(x_0))\| \cdot \|q(n, x_0, h)\| \tag{61}$$

$$= (\|D\eta(n, x'_n)\| + \|D\eta(n, x_n(x_0))\|) \cdot \|q(n, x_0, h)\| \tag{62}$$

$$\le \delta \|q(n, x_0, h)\|, \tag{63}$$

where $x'_n$ is a point on the line segment joining $x_n(x_0 + he_j)$ and $x_n(x_0)$. Since

$$\frac{\|\eta(i, x_i(x_0 + he_j)) - \eta(i, x_i(x_0))\|}{|h|} \le \alpha_i \epsilon$$

so $\|q(n, x_0, h)\|$ is bounded, denoted as

$$\|q(n, x_0, h)\| \leq M.$$

And then $\|\Delta_n\| \leq \delta M$. Define the operator as

$$\psi(k+1, x_0) = A(k, 0)P^+ e_j + \sum_{i=0}^{k} A(k, i+1)P^+ D\eta(i, x_i(x_0))\psi(i, x_0) \tag{64}$$

$$- \sum_{i=0}^{\infty} A(k+1+i, k+1)^{-1} P^- D\eta(k+1+i, x_{k+1+i}(x_0))\psi(k+1+i, x_0). \tag{65}$$

Consider the difference

$$q - \psi \tag{66}$$

$$= \sum_{i=0}^{k} A(k, i+1)P^+ \left(D\eta(i, x_i(x_0))\left(q(i, x_0, h) - \psi(i, x_0)\right) + \Delta_i\right) \tag{67}$$

$$- \sum_{i=0}^{\infty} A(k+1+i, k+1)^{-1} P^- \left(D\eta(k+1+i, x_{k+1+i}(x_0))\left(q(k+1+i, x_0, h) - \psi(k+1+i, x_0)\right) + \Delta_{k+1+i}\right). \tag{68}$$

Notice that the part of infinite sum converges to 0 as $k \to \infty$, one can choose $k$ large enough so that the norm of the infinite sum to be small, and then we have for any small $\epsilon' > 0$, the $\sup \|q - \psi\|$ satisfies

$$\sup \|q - \psi\| \leq \sum_{i=0}^{k} \left\| A(k, i+1)P^+ \left(D\eta(i, x_i(x_0))\left(q(i, x_0, h) - \psi(i, x_0)\right) + \Delta_i\right) \right\| + \epsilon' \tag{69}$$

$$\leq K'\epsilon d \sup \|q - \psi\| + K''\delta. \tag{70}$$

Where $K''$ is the bound from that $|\Delta_i| \to 0$ as $i \to \infty$. One choose neighborhood small enough so that $K'\epsilon d < \frac{1}{2}$ and then we have

$$\sup \|q - \psi\| < K''\delta.$$

Since $\delta \to 0$ as $h \to 0$, so $\sup \|q - \psi\| \to 0$ as $h \to 0$. And this means that the partial derivative $\frac{\partial x_n}{\partial \xi_i}$ exists and equals to $\psi$.

In the end we prove that $\bigcap_{k=0}^{\infty} \tilde{g}^{-1}(k, 0, U) \subset V(0)$ and this can be done by contradiction. Assume that there is an initial point $x_0$ not in $V(0)$ that generates a sequence $\{x_k\}_{k\in\mathbb{N}}$ such that $x_k \in U$ as $k \to \infty$. Since $x_{k+1}^- = C_{k+1}x_k^- + \eta^-(k, x_k)$, we have that

$$\left\|x_{k+1}^-\right\| = \left\|C_{k+1}x_k^- + \eta^-(k, x_k)\right\| \geq \left| \left\|C_{k+1}x_k^-\right\| - \left\|\eta^-(k, x_k)\right\| \right|.$$

Since $\|\eta^-(k, x_k)\| \to 0$ as $k \to \infty$ due to $\alpha_k$, and $\left\|C_{k+1}x_k^-\right\| \to \infty$ as $k \to \infty$ by assumption that $x_k^-$ is bounded away from 0. But this contradicts to the assumption $x_{k+1}^-$ is bounded in $U$. The proof completes. $\qquad\square$

## C.2 Proof of Corollary 4.3

*Proof.* For each $\mathbf{x}^* \in \mathcal{A}^*$, there is an associated open neighborhood $U_{\mathbf{x}^*}$ promised by the Corollary 4.2. $\bigcup_{\mathbf{x}^* \in \mathcal{A}^*} U_{\mathbf{x}^*}$ forms an open cover, and since $\mathbb{R}^d$ (more generally, any manifold) is second-countable we can find a countable subcover, so that $\bigcup_{\mathbf{x}^* \in \mathcal{A}^*} U_{\mathbf{x}^*} = \bigcup_{i=1}^{\infty} U_{\mathbf{x}_i^*}$

By the definition of global stable set, we have

$$W^s(\mathcal{A}^*) = \{\mathbf{x}_0 : \lim_{k \to \infty} \tilde{g}(k, 0, \mathbf{x}_0) \in \mathcal{A}^*\}.$$

Fix a point $\mathbf{x}_0 \in W^s(\mathcal{A}^*)$. Since $\tilde{g}(k, 0, \mathbf{x}_0) \to \mathbf{x}^* \in \mathcal{A}^*$, there exists some nonnegative integer $T$ and all $t \geq T$, such that $\tilde{g}(t, 0, \mathbf{x}_0) \in \bigcup_{\mathbf{x}^* \in \mathcal{A}^*} U_{\mathbf{x}^*} = \bigcup_{i=1}^{\infty} U_{\mathbf{x}_i^*}$. So $\tilde{g}(t, 0, \mathbf{x}_0) \in U_{\mathbf{x}_i^*}$ for some $\mathbf{x}_i^* \in \mathcal{A}^*$ and all $t \geq T$. This is equivalent to $\tilde{g}(T + k, T, \tilde{g}(T, 0, \mathbf{x}_0)) \in U_{\mathbf{x}_i^*}$ for all

$k \geq 0$, and this implies that $\tilde{g}(T, 0, \mathbf{x}_0) \in \tilde{g}^{-1}(T + k, T, U_{\mathbf{x}_i^*})$ for all $k \geq 0$. And then we have $\tilde{g}(T, 0, \mathbf{x}_0) \in \bigcap_{k=0}^{\infty} \tilde{g}^{-1}(T + k, T, U_{\mathbf{x}_i^*})$. Denote $S_{i,T} := \bigcap_{k=0}^{\infty} \tilde{g}^{-1}(T + k, T, U_{\mathbf{x}_i^*})$ and the above relation is equivalent to $\mathbf{x}_0 \in \tilde{g}^{-1}(T, 0, S_{i,T})$. Take the union for all nonnegative integers $T$, we have $\mathbf{x}_0 \in \bigcup_{T=0}^{\infty} \tilde{g}^{-1}(T, 0, S_{i,T})$. And union for all $i$ we obtain that $\mathbf{x}_0 \in \bigcup_{i=1}^{\infty} \bigcup_{T=0}^{\infty} \tilde{g}^{-1}(T, 0, S_{i,T})$ implying that $W^s(\mathcal{A}^*) \subset \bigcup_{i=1}^{\infty} \bigcup_{T=0}^{\infty} \tilde{g}^{-1}(T, 0, S_{i,T})$. Since $S_{i,T} \subset W_n(\mathbf{x}^*)$ from Corollary 4.2, and $W_n(\mathbf{x}^*)$ has codimension at least 1. This implies that $S_{i,T}$ has measure 0. Since the image of set zero-measure set under diffeomorphism is of measure 0, and countable union of zero-measure sets is of measure 0, we obtain that $W^s(\mathcal{A}^*)$ is of measure 0. □

## C.3 Proof of Theorem 5.3

*Proof.* Let $U \subset \mathbb{R}^d$ be an open ball centering at $\mathbf{x}^*$, the Taylor expansion of $g(k, \mathbf{x})$ in $U \cap M$ is of the form

$$g(k, \mathbf{x}) = g(k, \mathbf{x}^*) + (Id_{T_{\mathbf{x}^*}M} - \alpha_k \nabla_R^2 \Phi(\mathbf{x}^*)^{-1} \nabla_R^2 f(\mathbf{x}^*))(\mathbf{x} - \mathbf{x}^*) + \theta(k, \mathbf{x}).$$

The fact that $g(k, \mathbf{x})$ satisfies the condition 1 of Corollary 4.2 follows from the proof of Proposition 10, [3], i.e. $\nabla_R^2 \Phi(\mathbf{x}^*)^{-1} \nabla_R^2 f(\mathbf{x}^*)$ is similar to a symmetric linear operator (so diagonalizable) with at least one negative eigenvalue.
Next, we verify that $g(k, \mathbf{x})$ satisfies the condition 2 of Corollary 4.2. From the Taylor expansion, we have

$$\theta(k, \mathbf{x}) = g(k, \mathbf{x}) - \mathbf{x}^* - (Id_{T_{\mathbf{x}^*}M} - \alpha_k \nabla_R^2 \Phi(\mathbf{x}^*)^{-1} \nabla_R^2 f(\mathbf{x}^*))(\mathbf{x} - \mathbf{x}^*),$$

and

$$D_{\mathbf{x}}\theta(k, \mathbf{x}) = -\alpha_k \nabla_R^2 \Phi(\mathbf{x})^{-1} \nabla_R^2 f(\mathbf{x}) + \alpha_k \nabla_R^2 \Phi(\mathbf{x}^*)^{-1} \nabla_R^2 f(\mathbf{x}^*).$$

By the continuity of $\nabla_R^2 f$ and $\nabla_R^2 \Phi(\mathbf{x})^{-1}$, the same argument as the proof of Theorem 5.1 implies that the condition 2 of Corollary 4.2 is satisfied. Combing Corollary 4.2 and Corollary 4.3, we conclude that the stable set of saddle points has Lebesgue measure zero. □

## C.4 Proof of Theorem 5.4

*Proof.* Different from the other First-order methods, the results is not a direct consequence of Corollary 4.3, but instead, we need to apply part of the proof of Theorem 4.1. It is still necessary to verify that the Taylor expansion of $g(k, \mathbf{x})$ at $\mathbf{x}^*$ satisfies condition 1 and 2 of Corollary 4.2. From the proof of Proposition 3, [3], $g(k, \mathbf{x}) + \alpha_k \nabla f(g(k, \mathbf{x})) = \mathbf{x}$. By implicit differentiation, $Dg(k, \mathbf{x}) + \alpha_k \nabla^2 f(g(k, \mathbf{x}))Dg(k, \mathbf{x}) = I$, and

$$Dg(k, \mathbf{x}) = (I + \alpha_k \nabla^2 f(g(k, \mathbf{x})))^{-1}.$$

At saddle point $\mathbf{x}^*$, $Dg(k, \mathbf{x}^*) = (I + \alpha_k \nabla^2 f(g(k, \mathbf{x}^*)))^{-1}$ that is diagonalizable since $\nabla^2 f(\mathbf{x}^*)$ is diagonalizable. Suppose under the matrix $Q$, $Q\nabla^2 f(\mathbf{x}^*)Q^{-1} = H$ that is diagonal. Then

$$Q(I + \alpha_k \nabla^2 f(\mathbf{x}^*))^{-1} Q^{-1} = \left( Q(I + \alpha_k \nabla^2 f(\mathbf{x}^*))Q^{-1} \right)^{-1} \tag{71}$$

$$= \left( I + \alpha_k Q\nabla^2 f(\mathbf{x}^*)Q^{-1} \right)^{-1} \tag{72}$$

$$= (I + \alpha_k H)^{-1} \tag{73}$$

$$= \text{diag}\{\frac{1}{1 + \alpha_k \lambda_i}\}, \tag{74}$$

where $\lambda_i$ are the eigenvalues of $H$. Notice that $\frac{1}{1+\alpha_k \lambda_i} = 1 - \frac{\alpha_k \lambda_i}{1+\alpha_k \lambda_i}$, the stable-unstable decomposition in the proof of Corollary 4.1 holds. Furthermore, since $\alpha_k \in \Omega\left(\frac{1}{k}\right)$, $\frac{\alpha_k \lambda_i}{1+\alpha_k \lambda_i}$ is also of $\Omega\left(\frac{1}{k}\right)$. To see this, we can assume $\alpha_k \lambda_i \geq \frac{1}{k-1} = \frac{1}{k} \cdot \frac{k}{k-1}$, and then $\frac{k-1}{k}\alpha_k \lambda_i \geq \frac{1}{k}$ or $\left(1 - \frac{1}{k}\right)\alpha_k \lambda_i \geq \frac{1}{k}$, and thus $\alpha_k \lambda_i \geq \frac{1}{k}(1 + \alpha_k \lambda_i)$, implying that $\frac{\alpha_k \lambda_i}{1+\alpha_k \lambda_i} \geq \frac{1}{k}$. So the proof for Lemma C.1 and Lemma C.2 holds for the existence of stable manifold of proximal point algorithm if condition 2 of Corollary 4.2 is satisfied. To verify this, we consider the Taylor expansion of $g(k, \mathbf{x})$ at $\mathbf{x}^*$ has the form of

$$g(k, \mathbf{x}) = g(k, \mathbf{x}^*) + D_{\mathbf{x}}g(k, \mathbf{x}^*)(\mathbf{x} - \mathbf{x}^*) + \theta(k, \mathbf{x})$$
$$= \mathbf{x}^* + (I + \alpha_k \nabla^2 f(g(k, \mathbf{x}^*)))^{-1}(\mathbf{x} - \mathbf{x}^*) + \theta(k, \mathbf{x}),$$

and thus
$$\theta(k, \mathbf{x}) = g(k, \mathbf{x}) - \mathbf{x}^* - (I + \alpha_k \nabla^2 f(g(k, \mathbf{x}^*)))^{-1}(\mathbf{x} - \mathbf{x}^*).$$
So the differential
$$D_\mathbf{x}\theta(k, \mathbf{x}) = (I + \alpha_k \nabla^2 f(g(k, \mathbf{x})))^{-1} - (I + \alpha_k \nabla^2 f(g(k, \mathbf{x}^*)))^{-1}.$$
Since $f$ is of $C^2$, $g(k, \mathbf{x})$ and $\nabla^2 f(\mathbf{x})$ are continuous, and then $\|\theta(k, \mathbf{x}) - \theta(k, \mathbf{y})\| \le \alpha_k \epsilon \|\mathbf{x} - \mathbf{y}\|$ follows from the same argument as the proof of Theorem 5.1. So the verification completes and by Corollary 4.2 and Corollary 4.3, we conclude that the stable set of strict saddle points is of Lebesgue measure zero. □

## C.5 Proof of Theorem 5.5

*Proof.* According to the proof of Proposition 8, [3], for $\mathbf{v} \in T_{\mathbf{x}^*} M$,
$$D_\mathbf{x} g(k, \mathbf{x}^*)\mathbf{v} = P_{T_{\mathbf{x}^*} M} \mathbf{v} - \alpha_k P_{T_{\mathbf{x}^*} M} D(P_{T_{\mathbf{x}^*} M} \nabla \bar{f})(\mathbf{x}^*)\mathbf{v}.$$
Let $\mathbf{x} - \mathbf{x}^* \in T_{\mathbf{x}^*} M$, the Taylor expansion in the tangent space can be written as
$$g(k, \mathbf{x}) = g(k, \mathbf{x}^*) + P_{T_{\mathbf{x}^*} M}(\mathbf{x} - \mathbf{x}^*) - \alpha_k P_{T_{\mathbf{x}^*} M} D(P_{T_{\mathbf{x}^*} M} \nabla \bar{f})(\mathbf{x}^*)(\mathbf{x} - \mathbf{x}^*) + \theta(k, \mathbf{x}).$$
Using equation 4, [1], $P_{T_\mathbf{x} M} D(P_{T_\mathbf{x} M} \nabla \bar{f})(\mathbf{x}) = \nabla_R^2 f(\mathbf{x})$, which is exactly the Riemannian Hessian, and thus it is diagonalizable. So this verifies the condition 1 of Corollary 4.2. Furthermore, the Taylor expansion gives
$$\theta(k, \mathbf{x}) = g(k, \mathbf{x}) - \mathbf{x}^* - P_{T_{\mathbf{x}^*} M}(\mathbf{x} - \mathbf{x}^*) + \alpha_k P_{T_{\mathbf{x}^*} M} D(P_{T_{\mathbf{x}^*} M} \nabla \bar{f})(\mathbf{x}^*)(\mathbf{x} - \mathbf{x}^*),$$
and then
$$D_\mathbf{x}\theta(k, \mathbf{x}) = D_\mathbf{x} g(k, \mathbf{x}) - P_{T_{\mathbf{x}^*} M} + \alpha_k P_{T_{\mathbf{x}^*} M} D(P_{T_{\mathbf{x}^*} M} \nabla \bar{f})(\mathbf{x}^*).$$
The continuity of $\nabla^2 f$ implies that for each $\epsilon > 0$, there exist neighborhood of $\mathbf{x}^*$, such that $\|D_\mathbf{x}\theta(k, \mathbf{x})\| \le \epsilon$. Apply the argument in the proof of Theorem 5.1 (Fundamental Theorem of Calculus and Cauchy-Schwarz inequality), we can conclude that condition 2 of Corollary 4.2 is satisfied. then combing with Corollary 4.3, we have that the stable set of strict saddle points has measure (induced by metric $R$) zero. □

## C.6 Proof of Theorem 5.6

*Proof.* Let $\mathbf{x}^* \in \mathcal{X}^*$, then $\nabla f(\mathbf{x}^*) = 0$, and $g(k, \mathbf{x}^*) = \mathbf{x}^*$. To show that $\mathbf{x}^*$ is unstable, consider the differential
$$D_\mathbf{x} g(k, \mathbf{x}) = I - \alpha_k D_\mathbf{x}\left((R^{ij}) \cdot \nabla f(\mathbf{x})\right),$$
where
$$D_\mathbf{x}\left((R^{ij}) \cdot \nabla f(\mathbf{x})\right) = \begin{bmatrix} \frac{\partial}{\partial x_1}(R^{1j}\frac{\partial f}{\partial x_j}) & \cdots & \frac{\partial}{\partial x_d}(R^{1j}\frac{\partial f}{\partial x_j}) \\ \vdots & & \vdots \\ \frac{\partial}{\partial x_1}(R^{dj}\frac{\partial f}{\partial x_j}) & \cdots & \frac{\partial}{\partial x_d}(R^{dj}\frac{\partial f}{\partial x_j}) \end{bmatrix} \tag{75}$$
$$= \begin{bmatrix} \frac{\partial R^{1j}}{\partial x_1}\frac{\partial f}{\partial x_j} + R^{1j}\frac{\partial^2 f}{\partial x_1 \partial x_j} & \cdots & \frac{\partial R^{1j}}{\partial x_d}\frac{\partial f}{\partial x_j} + R^{1j}\frac{\partial^2 f}{\partial x_d \partial x_j} \\ \vdots & & \vdots \\ \frac{\partial R^{dj}}{\partial x_1}\frac{\partial f}{\partial x_j} + R^{dj}\frac{\partial^2 f}{\partial x_1 \partial x_j} & \cdots & \frac{\partial R^{dj}}{\partial x_d}\frac{\partial f}{\partial x_j} + R^{dj}\frac{\partial^2 f}{\partial x_m \partial x_j} \end{bmatrix} \tag{76}$$
$$= \begin{bmatrix} \frac{\partial R^{1j}}{\partial x_1}\frac{\partial f}{\partial x_j} & \cdots & \frac{\partial R^{1j}}{\partial x_d}\frac{\partial f}{\partial x_j} \\ \vdots & & \vdots \\ \frac{\partial R^{dj}}{\partial x_1}\frac{\partial f}{\partial x_j} & \cdots & \frac{\partial R^{dj}}{\partial x_d}\frac{\partial f}{\partial x_j} \end{bmatrix} + \begin{bmatrix} R^{1j}\frac{\partial^2 f}{\partial x_1 \partial x_j} & \cdots & R^{1j}\frac{\partial^2 f}{\partial x_d \partial x_j} \\ \vdots & & \vdots \\ R^{dj}\frac{\partial^2 f}{\partial x_1 \partial x_j} & \cdots & R^{dj}\frac{\partial^2 f}{\partial x_d \partial x_j} \end{bmatrix}. \tag{77}$$
Since at $\mathbf{x}^*$, $\nabla f(\mathbf{x}^*) = 0$, i.e. $\frac{\partial f}{\partial x_j} = 0$ for all $j$, we have
$$D_\mathbf{x}\left((R^{ij}) \cdot \nabla f(\mathbf{x}^*)\right) = \begin{bmatrix} R^{1j}\frac{\partial^2 f}{\partial x_1 \partial x_j} & \cdots & R^{1j}\frac{\partial^2 f}{\partial x_d \partial x_j} \\ \vdots & & \vdots \\ R^{dj}\frac{\partial^2 f}{\partial x_1 \partial x_j} & \cdots & R^{dj}\frac{\partial^2 f}{\partial x_d \partial x_j} \end{bmatrix}_{\mathbf{x}=\mathbf{x}^*} = (R^{ij}) \cdot \left(\frac{\partial^2 f}{\partial x_i \partial x_j}\right)\bigg|_{\mathbf{x}=\mathbf{x}^*}.$$

Recall that $\left(R^{ij}\right) = \left(R_{ij}\right)^{-1}$, and as it is shown in [3], by the similar transformation under $(R_{ij})^{\frac{1}{2}}$, we have

$$(R_{ij})^{\frac{1}{2}} \cdot D_{\mathbf{x}}\left(\left(R^{ij}\right) \cdot \nabla f(\mathbf{x}^*)\right) \cdot (R_{ij})^{-\frac{1}{2}} = (R_{ij})^{-\frac{1}{2}} \cdot \left(\frac{\partial^2 f}{\partial x_i \partial x_j}\right) \cdot (R_{ij})^{-\frac{1}{2}},$$

that is a symmetric matrix, so it is diagonalizable. And thus, $D_{\mathbf{x}}\left(\left(R^{ij}\right) \cdot \nabla f(\mathbf{x}^*)\right)$ is diagonalizable and has the same eigenvalue with $\left(\frac{\partial^2 f}{\partial x_i \partial x_j}\right)$, meaning that it has negative eigenvalues. So the Riemmanian Hessian at $\mathbf{x}^*$ as at least one negative eigenvalue.

The Taylor expansion of $g(k, \mathbf{x})$ at $\mathbf{x}^*$ is

$$g(k, \mathbf{x}) = g(k, \mathbf{x}^*) + \left(I - \alpha_k \left(R^{ij}(\mathbf{x}^*)\right) \cdot \left(\frac{\partial^2 f}{\partial x_i \partial x_j}(\mathbf{x}^*)\right)\right)(\mathbf{x} - \mathbf{x}^*) + \theta(k, \mathbf{x}),$$

where

$$\left(R^{ij}(\mathbf{x}^*)\right) \cdot \left(\frac{\partial^2 f}{\partial x_i \partial x_j}(\mathbf{x}^*)\right) = \left(R^{ij}\right) \cdot \left(\frac{\partial^2 f}{\partial x_i \partial x_j}\right)\bigg|_{\mathbf{x}=\mathbf{x}^*}.$$

We have

$$\theta(k, \mathbf{x}) = g(k, \mathbf{x}) - \mathbf{x}^* - \left(I - \alpha_k \left(R^{ij}(\mathbf{x}^*)\right) \cdot \left(\frac{\partial^2 f}{\partial x_i \partial x_j}(\mathbf{x}^*)\right)\right)(\mathbf{x} - \mathbf{x}^*)$$

and

$$D_{\mathbf{x}}\theta(k, \mathbf{x}) = -\alpha_k \left(R^{ij}(\mathbf{x})\right) \cdot \left(\frac{\partial^2 f}{\partial x_i \partial x_j}(\mathbf{x})\right) + \alpha_k \left(R^{ij}(\mathbf{x}^*)\right) \cdot \left(\frac{\partial^2 f}{\partial x_i \partial x_j}(\mathbf{x}^*)\right).$$

By the continuity of $\left(\frac{\partial^2 f}{\partial x_i \partial x_j}(\mathbf{x})\right)$, the same argument as the verification of condition 2 in the proof of Theorem 5.1 implies that $\theta(k, \mathbf{x})$ satisfies the condition 2 of Corollary 4.2. Combining with Corollary 4.3, we conclude that the stable set of strict saddle points has measure (induced by metric $R$) zero. $\qquad\square$

## Footnotes

[1]Jacobian of function $g$.