[Reviews · NeurIPS 2019]

Reviewer 1



While the result might be considered incremental, extending recent work by considering a more sophisticated class of algorithms, the paper still seems quite relevant. This appears to be a good next step in an overarching research project of bringing important results from dynamical systems theory to the context of non-convex optimization techniques applied to machine learning problems, what speaks for the originality of the work. Its significance comes from the considered extension being relevant and not trivial. The paper depends on knowledge that many readers may not be familiar with, and for that reason can be a little difficult to follow. Bringing this knowledge to the field is precisely one of the best aspects of the paper, though. In many places it is quite intructional and easy to follow. In others not quite so. It is hard to suggest improvements, though. The authors cannot be asked to provide a complete introductory text on the required subjects. While that is clear, the text seems intended to different audiences in different parts, and could perhaps be made more homogeneous in that regard, improving its quality and clarity.

Reviewer 2



This paper is well written and well argued. It is likely to be of wide interest to the community. The paper is quite technical, although, for the results they are trying to establish, this is necessary.

Reviewer 3



In this paper, the authors prove that first order methods with vanishing step sizes can also almost always avoid saddle points. The step-size decay is time dependent, the resulting analysis for the main theorem is a non-trivial extension of the proof for constant step sizes. Post rebuttal: Classical stochastic approximation proves require the sum of step-sizes to go to infinity while the sum of squares of step-sizes to be less than infinity. Since, the latter condition is no longer required, the gap between the allowed step-sizes for stochastic approximation and what is proposed by this paper is unclear. Do the additional conditions (Lines 167-168) lead to the elimination of the condition? 1) The motivation for first order methods with decaying step sizes over constant step sizes is not clear. If methods like stochastic gradient descent are used as motivation in the introduction, then its surprising that results will not hold for any first order method with noisy estimates for gradients like SGD. 2) The rebuttal is not convincing enough in how exactly Line 54 implies the elimination of the condition. Moreover, it is obvious why block co-ordinate descent fails since it specifically takes in gradient estimates with non-zero noise and sum of squares of step-sizes is infinite. This will be the same issue the authors run into when proving results for something like SGD or any stochastic approximation based methods. However, the paper still presents non-trivial breakthrough for vanishing step-sizes using dynamical systems theory which can be viewed as a good first step.

[Author Response · NeurIPS 2019]

We sincerely thank all the reviewers for their meticulous work and their helpful and detailed comments. Specifically, thank you for your suggestions on expanding upon our current figure/simulations as well as adding more intuition and high level ideas so as to help the reader through the admittedly complex proof. We will gladly incorporate them!

**[To Reviewer #1:]** Thank you very much for your hard work and your supportive comments for our paper!

**1.**"perhaps between section 3 and 4 . . . what would be great." Thank you for this suggestion. We will update the revised version analogously.

**2.**"Regardless,... supplementary article." In section 14.4.2 titled "Variable Step Size", the step-size $\eta_t = \frac{B}{\rho\sqrt{t}}$ is used in SGD. For the Stable manifold theorem in line 30, we refer to Chapter 5 Theorem III.7 in "Global Stability of Dynamical Systems" by M. Shub and in line 130, we refer to Section 2.7 in "Differential Equations and Dynamical Systems" by L. Perko.

**3.**"Might be interesting to add a note... Is this correct?" Yes, for the constant step-size $\alpha$, we need $\alpha$ to be less than the inverse of largest eigenvalue ([10], Lee et al, 2019). In the case of vanishing step-sizes the upper bound is unnecessary from an asymptotic perspective, but it is practically better to start with $\alpha_0 < 1/L$.

**4.**"Does the paper... stuck in a saddle point" The last parts of the statements of Theorem 4.1 and Corollary 4.2 assert that almost all points in a neighborhood of a saddle point will be transported out of the neighborhood in a finite number of steps. The convergence criterion "$f(x_{k+1}) \leq f(x_k) - \alpha_k(1 - \frac{\alpha_k L}{2})||\nabla f(x_k)||^2$ for all $k$" for convex functions can be used in neighborhoods of local minima (e.g.p126, *Convex Optimization Algorithms*, by Bertsekas). It asserts that once a point enters a neighborhood of local minimum, it converges. So convergence can be attained by combining these results.

**5.**"This is of course... that area of study." Recent progress (e.g.[6] and Jin et al 2017) show that perturbed gradient descent can escape saddles efficiently and then find approximate local minima in polynomial time. The efficiency of deterministic methods relies heavily on the specific structure of functions and initialization. In "GD can take exponential time...", Du et al constructed a function for which GD is significantly slowed down by saddles. But GD works well for the functions of matrix factorization(Jain et al 2017), phase retrieval, dictionary learning and so on (pointed in [10]). Generic efficiency results for first order methods without noise is an interesting direction for future research. Especially for variable step-sizes, very little is known. Methods using information of Hessian and curvature are studied in recent references (e.g."A geometric analysis of phase retrieval" Sun et al and "On noisy negative curvature descent" Liu and Yang, "A generic approach..." by Reddi et al, 2017). We will expand upon these connections in the revision.

**[To Reviewer #2:]** Thank you very much for your hard work and your supportive comments for our paper!

"If it was a space issue..." The case of coordinate descent is left open due to technical difficulties. Our main result (Theorem 1.1) relies on the fact that the differential of update rules can be written as $I - \alpha_k H$ where $H$ has no complex eigenvalues. The differential of coordinate descent can lead to complex eigenvalues, and thus our stable manifold theorem (Theorem 4.1) cannot be applied as is. Actually the issue is even more complicated. Suppose we had a generalization of Theorem 4.1 that addresses complex eigenvalues, it is not clear if it would imply the desired result for coordinate descent, because the differential of coordinate descent update rule cannot be written in the form of $I - \alpha_k H$ with $H$ diagonalizable. New ideas will probably be required to address these intricate issues. We will make a formal remark about this to hopefully stimulate future research on this important direction.

**[To Reviewer #3:]** Thank you for your work, comments and suggestion on improvements.

**1.**"Since, the latter condition is no longer required, ... unclear." The condition of $\sum \alpha_k^2 < \infty$ is unnecessary in the deterministic case. It can be understood in the following sense:

*Intuition:* 1. The reason to introduce $\sum \alpha_k^2 < \infty$ is to make the total variance of the increments finite (e.g. p699 of [16] R.Pemantle,1990). But deterministic methods can be seen as stochastic methods with noise identically 0, so the "total variance" is identically 0 (which is finite) for any step-sizes. 2. Consider the case when the step-size is a constant $\alpha$ (then $\sum \alpha^2 = \infty$). We already know that our result is true in this case since this is exactly the result of [10] Lee et al, 2019. So the condition $\sum \alpha_k^2 = \infty$ should not cause any obstruction in our deterministic setting.

*Technical sketch:* Formally, the proofs of Theorem 4.1 and Corollary 4.2 do not require $\sum \alpha_k^2 < \infty$. Briefly the proof is to use the Banach Fixed Point Theorem to show the existence and uniqueness of the stable manifold. The place where property of $\alpha_k$ is actually used is in the proofs of Lemma C.1 and Lemma C.2, where we show that the sums $R_k = \sum_{i=0}^{\infty} \alpha_{k+1+i} \prod_{j=k+1}^{k+1+i}(1 - \alpha_j \lambda)^{-1}$ with $\lambda < 0$, and $S_k = \alpha_k + \sum_{i=0}^{k-1} \alpha_i \prod_{j=i+1}^{k}(1 - \alpha_j \lambda)$ with $\lambda > 0$ are bounded as $k \to \infty$. To see that the restriction of $\sum \alpha_k^2 < \infty$ is unnecessary, consider the case of constant step-size (which trivially implies $\sum \alpha_k^2 = \infty$), then $R_k$ and $S_k$ become geometric series with ratios less than 1, so boundedness immediately follows.

**2.** "Do the additional conditions (lines 167-168) lead to elimination of the condition?" That is correct. These conditions are used to obtain the expression in Line 54 in the supplementary material.

[Meta-Review · NeurIPS 2019]

Dear Authors, Thank you for your submission to NeurIPS. Your work has generated fruitful and interesting discussions, where both pros and cons about your proposal were raised. The reviewers consent for acceptance. Two points to be considered before the final version of the paper: 1. Please elaborate more on Line 54 implies elimination of the condition of sum of squares of step sizes less than infinity. 2. In the conclusions section, it is clear the statement for co-ordinate descent is true; the proof for co-ordinate descent is still an open question. We encourage the authors to restate the conclusions, as reviewers are concerned that the proof breaks down for any first order method which has noisy estimate of gradients like block co-ordinate descent, SGD, etc.